# Plug, Play, and Generalize: Length Extrapolation with Pointer-Augmented Neural Memory

**Hung Le**                                                                              *thai.le@deakin.edu.au*
*Applied AI Institute*
*Deakin University, Australia*

**Dung Nguyen**                                                                   *dung.nguyen@deakin.edu.au*
*Applied AI Institute*
*Deakin University, Australia*

**Kien Do**                                                                                   *k.do@deakin.edu.au*
*Applied AI Institute*
*Deakin University, Australia*

**Svetha Venkatesh**                                                  *svetha.venkatesh@deakin.edu.au*
*Applied AI Institute*
*Deakin University, Australia*

**Truyen Tran**                                                                 *truyen.tran@deakin.edu.au*
*Applied AI Institute*
*Deakin University, Australia*

**Reviewed on OpenReview:** *https://openreview.net/forum?id=dyQ9vFbF6D*

## Abstract

We introduce Pointer-Augmented Neural Memory (PANM), a versatile module designed to enhance neural networks' ability to process symbols and extend their capabilities to longer data sequences. PANM integrates an external neural memory utilizing novel physical addresses and pointer manipulation techniques, emulating human and computer-like symbol processing abilities. PANM facilitates operations like pointer assignment, dereferencing, and arithmetic by explicitly employing physical pointers for memory access. This module can be trained end-to-end on sequence data, empowering various sequential models, from simple recurrent networks to large language models (LLMs). Our experiments showcase PANM's exceptional length extrapolation capabilities and its enhancement of recurrent neural networks in symbol processing tasks, including algorithmic reasoning and Dyck language recognition. PANM enables Transformers to achieve up to 100% generalization accuracy in compositional learning tasks and significantly improves performance in mathematical reasoning, question answering, and machine translation. Notably, the generalization effectiveness scales with stronger backbone models, as evidenced by substantial performance gains when we test LLMs finetuned with PANM for tasks up to 10-100 times longer than the training data.

## 1 Introduction

Systematic generalization underpins intelligence, and it relies on the ability to recognize abstract rules, extrapolating them to novel contexts that are distinct yet semantically similar to the seen data. Current neural networks or statistical machine learning fall short of handling novel data generated by symbolic rules even though they have achieved state-of-the-art results in various domains. Some approaches can show decent generalization for single or set input data (Bahdanau et al., 2018; Gao et al., 2020; Webb et al., 2020). Yet, neural networks in general still fail in sequential symbol processing tasks, even with slight

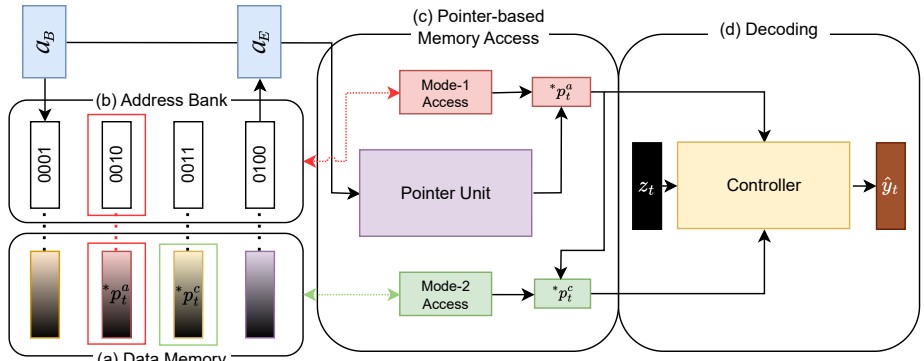

Figure 1: PANM architecture. (a) The data memory contains the encoded input sequence (b) The address bank contains physical addresses associated with data memory slots. The base and end addresses $(a_B, a_E)$ define the address range of the input sequence. (c) The Pointer Unit takes $a_B, a_E$, recurrently generates the current pointer $p_t^a$ and gets its value ${}^*p_t^a$ via Mode-1 (red)/2 (green) Access. (d) The Controller takes pointer information, decoding input ($z_t = y_{t-}$), and produce the $t$-th output token $\hat{y}_t$. We note that for illustration purposes, the figure depicts a special case where the pointer perfectly matches an address (e.g., $p_t^a = 0010$), in practice, the pointer may not point exactly to a single address.

novelty during inference (Lake & Baroni, 2018; Delétang et al., 2022). For instance, these models can easily learn to duplicate sequences of 10 items, but they will fail to copy sequences of 20 items if they were not part of the training data. These models overfit the training data and perform poorly on out-of-distribution samples such as sequences of greater length or sequences with novel compositions. The issue also affects big models like Large Language Models, making them struggle with symbolic manipulation tasks (Qian et al., 2023). *This indicates that current methods lack a principled mechanism for systematic generalization.*

From a neuroscience perspective, it has been suggested that the brain can execute symbol processing through variable binding and neural pointers, wherein the sensory data are conceptualized into symbols that can be assigned arbitrary values (Kriete et al., 2013). Like the brain, computer programs excel at symbolic computations. Programmers use address pointers to dynamically access data or programs, and have flexible control over the variable. Their programs can work appropriately with unseen inputs.

Building on these insights, we propose a pointer-based mechanism to enhance generalization to unseen length in sequence prediction, which is a crucial problem that unifies all computable problems (Solomonoff, 2010). *Our mechanism is based on two principles:* **(I)** explicitly modeling pointers as physical addresses, and **(II)** strictly isolating pointer manipulation from input data. As such, we need to design a memory that supports physical pointers, and create a model that manipulates the pointers to perform abstract rules and access to the memory. Our memory, dubbed Pointer-Augmented Neural Memory (PANM), is slot-based RAM (Von Neumann, 1993) where each memory slot consists of two components: data and address. Unlike initial endeavors that implicitly model pointers as attention softmax (Vinyals et al., 2015; Kurach et al., 2015), our addresses are generated to explicitly simulate physical memory addresses, i.e., incremental binary numbers, which is critical for generalization to longer sequences.

To manipulate a pointer, we create an address bank that contains physical addresses corresponding to the input sequence, and use a neural network called Pointer Unit that is responsible for transforming pointers from an initial address in the address bank. Through attention to the address bank, a new pointer is generated as a mixture of the physical addresses, which can point to different memory slots to follow the logic of the task. We aim to let the Pointer Unit learn the symbolic rules of the task in an end-to-end manner. Finally, given a (manipulated) pointer, the model can access the data through 2 modes of pointer-based access: pointer dereference (Mode-1) and relational access (Mode-2). Our memory can be plugged into common encoder-decoder backbones such as LSTM or Transformer.

Our contribution is a novel memory architecture, PANM, which incorporates explicit pointer and symbol processing, seamlessly enhancing sequential models for better generalization. In our experiments, we inte-

grate PANM into a range of deep learning backbone models of varying scales to assess the improvement in generalization:

- First, we add PANM to recurrent neural networks (LSTM) and memory-augmented neural networks (SRNN), demonstrating excellent generalization in symbol-processing tasks like algorithms and context-free grammar.

- We also apply PANM to Transformer models, improving their performance on compositional learning with SCAN and mathematics datasets. Additionally, PANM significantly enhances Transformer and BERT generalization in question answering and machine translation tasks.

- Finally, we demonstrate PANM's scalability by integrating it with LLMs (Llama2-7B), showing significant improvement in length extrapolation for NLP tasks requiring symbolic reasoning.

Our focus is not on striving for state-of-the-art results requiring specialized designs tailored to specific tasks. Instead, we aim to highlight that *universal generalization improvement can be achieved by integrating our memory module into various sequential models, with minimal architectural changes, and showcase the importance of using fundamental generalizing principles to address limitations of current deep learning.*

## 2 Methods

### 2.1 Problem Formulation

In sequence-to-sequence (s2s) problems, each data point is an input sequence $X_i = \left\{x_t^i\right\}_{t=1}^{l(X_i)}$, associated with a target sequence $Y_i = \left\{y_t^i\right\}_{t=1}^{l(Y_i)}$ where $l$ is a function returning the length of the sequence. A model $\Phi$ takes the input $X_i$ and generates an output sequence $\hat{Y}_i = \left\{\hat{y}_t^i\right\}_{t=1}^{l(\hat{Y}_i)}$ where the predicted sequence terminates as the model outputs token $\hat{y}_{t=l(\hat{Y}_i)}^i = \text{EOS}$ where EOS is a special end-of-sequence token. Each predicted token is sampled from a categorical distribution, parameterized by $\Phi$ and conditioned on the input sequence and optionally with previous output tokens: $\hat{y}_t^i \sim p_\Phi(y_t|X_i, y_{t-}^i)$ where $y_{t-}^i$ can be $\left\{y_k^i\right\}_{k=1}^{t-1}$ (true outputs) or $\left\{\hat{y}_k^i\right\}_{k=1}^{t-1}$ (predicted outputs) or even zero, depending on the setting (training or inference). We train $\Phi$ by minimizing the cross-entropy loss:

$$\mathcal{L} = \mathbb{E}_i \left[ -\sum_t \log p_\Phi(y_t^i|X_i, y_{t-}^i) \right]$$

We are interested in the ability to handle inputs of arbitrary length, so we focus on settings in which the length of testing input sequences is larger than that of training ones: $\max l(X_i) < \min l(X_j)$ with $X_i \in \mathcal{D}_{train}$ and $X_j \in \mathcal{D}_{test}$. In the following sections, when there is no confusion, we will drop the sample index $i$ or $j$ for ease of reading. We note that autoregression is a special case of the s2s formulation where the input and output are from the same domain, and the target sequence is one step ahead of the input sequence.

### 2.2 Pointer Modeling

Computers are powerful thanks to their ability to manipulate pointers. These pointers store the address of data in memory. Following $\boldsymbol{C}$ programming language notation, let $p$ denote a pointer associated with a data $d$ in memory M, then $p$ is also the address of the memory slot containing $d$, i.e., $p = \&d$ where $\&$ operator returns the pointer associated with value $d$. We can access the data pointed by $p$ as $^*p = d$ where $*$ operator returns the memory value associated with pointer $p$, which is also known as *pointer dereference.*

We can manipulate the pointer to execute various tasks. For example, given a list $X$ storing elements in consecutive memory slots, $\&X$ denotes the pointer of the first element of the list. If the task is to copy the list $X$ to a new list $Y$, using pointers, this task can be executed by iterating over the elements of the list and copying each element from $X$ to $Y$ regardless of the list length and the values in $X$. The copying

process can be described as follows: (1, *assignment*) Initialize 2 pointers pointing at the first element of $X$ and $Y$, respectively: $p_X = \&X; p_Y = \&Y$; (2, *dereference*) Access the value pointed to by $p_X$ (i.e., the current element of $X$) and copy it to the location pointed to by $p_Y$ (i.e., the corresponding position in $Y$): ${}^*p_Y = {}^*p_X$; (3, *arithmetic*) Increment both pointers to move to the next elements in their respective lists, $p_X = p_X + 1; p_Y = p_Y + 1$. Repeat this procedure until all elements in $X$ have been copied to $Y$.

In this paper, we propose a way to model pointers by constructing a bank of addresses, analogous to the addresses in computer architecture. The address bank starts with a *base address* $a_B$ and increases to form an arithmetic sequence with the common difference of 1. For example, if the bank has 3 addresses and $a_B = 3$, the addresses are $\texttt{A} = \{3, 4, 5\}$. We represent the address as $b$-bit binary vectors, so we have $\texttt{A} = \{0010, 0011, 0100\}$ when $b = 4$. The address space is $2^b$ $b$-bit binary vectors. Given a memory $\texttt{M}$ containing $l(\texttt{M})$ slots, we bind each slot to an address in the address bank $\texttt{A}$ such that $\texttt{A}[t] = \&\texttt{M}[t]$ and ${}^*\texttt{A}[t] = \texttt{M}[t]$ $(1 \leq t \leq l(\texttt{M}))$. We use $a_E$ as the *end address* to refer to the final address corresponding to the last item in $\texttt{M}$.

The memory $\texttt{M}$ stores the input sequence, and its size depends on the input length: $l(\texttt{M}) = l(X)$. To enable generalization, the address space should cover a wide range of addresses that is greater than the sequence length range (i.e., $2^b > \max l(X)$). More importantly, during training, all possible addresses should be exposed to the model. Otherwise, any unexposed address will confuse the model when it appears during inference. As such, during training, we *uniformly sample the base address $a_B$* from the address space to construct the address bank $\texttt{A}$ for each sequence memory $\texttt{M}$. This ensures any address in the address space will eventually appear between the base and end addresses. See Appendix B for an illustrative example of the base address sampling mechanism and its complexity.

Given the address bank, we can perform pointer-based procedures to achieve generalization. To do so, we need pointer variables $p_t$ denoting pointers used by the model at timestep $t$. As for the copy task, the model outputs correctly by accessing and manipulating the pointer variables through 3 important pointer operations: $p_t = \texttt{A}[t]$ (*assignment*); $\hat{y}_t^i = {}^*p_t$ (*dereference*); $p_t = p_t + 1$ (*arithmetic*), which will be described in the next section.

### 2.3 Pointer-Augmented Neural Memory (PANM)

PANM acts as an external memory module for any neural network to support it handling sequence data. In such a memory-augmented neural network, a neural Controller ($\texttt{Ctrl}$) interacts with the memory ($\texttt{M}$) to read/write data and make predictions on the output target. Unlike traditional neural memory, PANM is equipped with an address bank ($\texttt{A}$) and a Pointer Unit ($\texttt{PU}$) to support pointer operations. To simplify memory writing operations, PANM transforms the whole input sequence $X$ to the memory $\texttt{M}$ in the encoding phase such that $L = l(\texttt{M}) = l(X)$ using $\texttt{M} = \texttt{Encoder}_\theta(X)$ where $X \in \mathbb{R}^{d_x \times L}$, $\texttt{M} \in \mathbb{R}^{d_m \times L}$ and the $\texttt{Encoder}$, parameterized by $\theta$, can be any neural encoder such as LSTM or Transformer. The address bank $\texttt{A} \in \{0, 1\}^{b \times L}$ is then created and bound to $\texttt{M}$ as mentioned in the previous section. During decoding, the encoded information in $\texttt{M}$ is not changed and the controller focuses only on reading from $\texttt{M}$ to produce the right output sequence. An overview of PANM decoding process is given in Fig. 1.

### 2.3.1 Pointer Unit

At each timestep $t$ of the decoding process, PANM makes use of pointer variables $p_t^a$, which are initialized as a valid address in the address space and then updated by the Pointer Unit $\texttt{PU}$. In tasks like summarizing a lengthy article by extracting key points from specific paragraphs, the Pointer Unit functions like a sophisticated "bookmark" system, starting at the targeted paragraph and sequentially progressing through each key point, effectively managing documents of any length by using relative positions. In particular, the $\texttt{PU}$, implemented as an GRU (Chung et al., 2014), takes an address from $\texttt{A}$ as its initial inputs, e.g., $p_0^a = a_B$, and recurrently produces a key $h_t^a$ that performs *address attention* to create succeeding pointer variables $p_t^a$:

$$h_t^a = \text{GRU}_\varphi \left( p_{t-1}^a, h_{t-1}^a \right) \tag{1}$$

$$w_t^a[n] = \text{softmax} \left( \frac{h_t^a g_\varphi^a (\text{A}[n])}{\|h_t^a\| \left\| g_\varphi^a (\text{A}[n]) \right\|} \right) \tag{2}$$

$$p_t^a = \text{A} w_t^a \tag{3}$$

where $h_0^a$ is initialized as $\overrightarrow{0}$ in Eq. (1). In Eq. (2), $1 \leq n \leq l(X)$, $\varphi$ denotes the parameters of the PU and $g^a(\cdot)$ is a feed-forward neural network to transform the address to the same space as $h_t^a$. According to § 1's principle **I**, $p_t^a$ is "softly" *assigned* a physical address value in the address bank. Our pointer, $p_t^a$, offers several advantages over "implicit pointers" made up of the attention weights ($w_t^a$), which are commonly utilized in previous works (Vinyals et al., 2015; Luong et al., 2015). First, $p_t^a$ is a combination of physical addresses represented by binary numbers, and therefore its dimension is generally independent of the sequence length. In contrast, the dimension of $w_t^a$ varies with the input length. Therefore, arithmetic transformations on $p_t^a$ are easier than on $w_t^a$. Second, longer testing length poses challenges for traditional attentions to accurately produce $w_t^a$ pointing to unseen location. Using "physical key" $A$ to compute $w_t^a$ mitigates this issue by employing random physical address ranges (see § 2.2).

Following § 1's principle **II**, the PU *recurrently transforms* the original $p_0^a$ to a series of pointers $\{p_t^a\}_{t=1}^{l(\hat{Y})}$ suitable for the current task *without using input data*. This prevents unseen testing inputs disturb PU's transformations. In the copy example, an ideal arithmetic transformation ensure $p_0^a = a_B$ and $p_{t+1}^a = p_t^a + 1$, which performs perfectly for any sequence whose length $\leq 2^b$. We aim to learn PU to automatically discover pointer manipulation rules from the task. As the rules are learned, generalization is achieved even when the testing sequence is longer or contains novel items.

### 2.3.2 Pointer-based Addressing Modes

**Mode 1** In this mode, PANM's pointer functions like an index, much like accessing a particular row in a data table or a sentence in a passage, so if the PU points to a sentence, it directly retrieves that sentence, regardless of the passage content. Particularly, the content from memory M is retrieved directly by dereferencing pointers. To *dereference* pointer $p_t^a$, we utilize the $A - M$ binding and the address attention weight $w_t^a$, retrieving the pointer value associated with $p_t^a$ as $^*p_t^a = M w_t^a$. Through this dereference operator, we can access to arbitrary data in the memory M without relying on the content of the memory. This property enables robustness in handling new sequence when the memory content is novel and the process stays the same (e.g., copy a never-seen-before sequence). Accessing M indirectly via A allows more memory slots to be added to M during inference without affecting the processing rule as long as the PU can transform the pointer variable correctly. During training, PU experiences pointer variables covering the whole address space because the base address is sampled randomly. Hence, it is possible for PU to correctly transform pointers that point to extended addresses of a growing memory as long as the pointer manipulation rule does not change. The address attention can be used for multiple pointer variables. In this case, there would be multiple pointer units $\{\text{PU}_h\}_{h=1}^{H^a}$ responsible for several $\left\{ p_{t,h}^a \right\}_{h=1}^{H^a}$ and $\left\{ ^*p_{t,h}^a \right\}_{h=1}^{H^a}$ where $H^a$ is the number of attention heads. These pointer values will be used by the Controller for other memory reading.

**Mode 2** This mode uses a more complicated memory access mechanism to capture relations between pointers in complicated reasoning tasks. *The accessed content is not the one associated with the current pointer*, but those whose contents are related to the current pointer's value. As an example, selection sort algorithm may require comparing items in a list with the Mode-1 pointer's item to select the greater one. Another example is to summarize a document by identifying the most important sentences based on their relevance to a query. In this mode, PANM's pointer goes beyond direct retrieval by using the content of a sentence to generate a query that ranks sentences based on relevance, much like running a search query to identify and select the most important sentences in a document based on their contextual significance. We simulate that using attention with the query as the current pointer value:

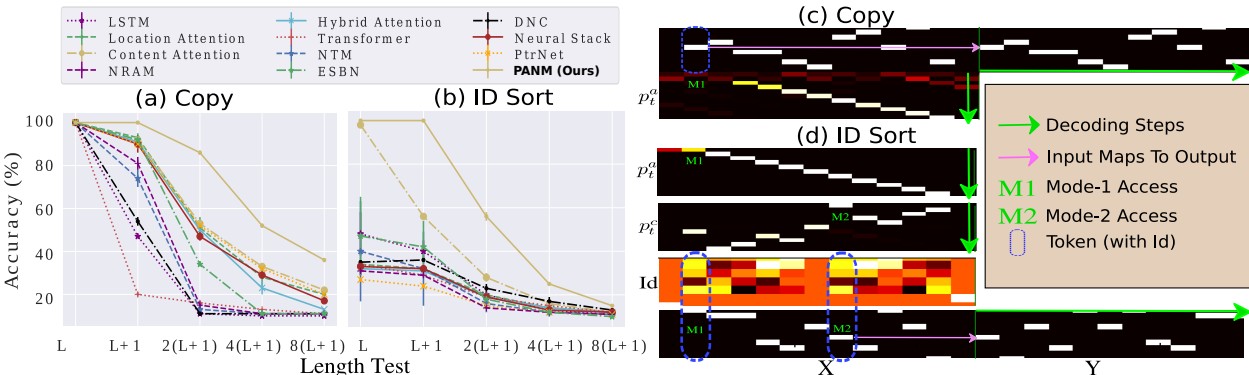

Figure 2: Exemplar results on 2 algorithms. (a, b) Test accuracy (mean $\pm$ std) over 5 runs on Copy and ID Sort on each length test, respectively. Random predictor would reach around 10% accuracy. (c,d) Visualization of data and pointer's slots for Copy and ID Sort, respectively.

$$q_t = g^c_\varphi \left( \left\{ {}^* p^a_{t,h} \right\}^{H^a}_{h=1} \right) ; \tag{4}$$

$$w^c_t [n] = \text{softmax} \left( \frac{q_t \mathtt{M}[n]}{\|q_t\| \, \|\mathtt{M}[n]\|} \right) \tag{5}$$

$$^* p^c_t = \mathtt{M} w^c_t \tag{6}$$

Here, the pointer attention takes the concatenated values $\left\{ {}^* p^a_{t,h} \right\}^{H^a}_{h=1}$ as input, transforms them to a query $q_t$ using a feed-forward neural network $g^c(\cdot)$, and returns the related pointer value $^* p^c_t$ through attention mechanisms on $\mathtt{M}$. Intuitively, the Pointer Unit manipulates the Mode-1 pointer $p^a_t$ such that it retrieves the desired content pointed by the Mode-2 pointer $p^c_t$. We can also have multi-head attention, which results in $\left\{ {}^* p^c_{t,h} \right\}^{H^c}_{h=1}$ where $H^c$ is the number of attention heads.

### 2.3.3 The Controller

The Controller $\mathtt{Ctrl}$ is responsible for decoding the memory to produce outputs. Unlike other methods, we have pointer-based memory access to provide the controller with symbol-processing information. In particular, at the $t$-th step of the decoding, $\mathtt{Ctrl}$ takes the pointer values (mode 1 and 2) as input together with an optional decoding input ($z_t = y_{t-}$), and uses a GRU to recurrently produce the hidden state $h^c_t$:

$$h^c_t = \mathtt{GRU}_\lambda \left( \left[ \left\{ {}^* p^a_{t,h} \right\}^{H_a}_{h=1}, \left\{ {}^* p^c_{t,h} \right\}^{H_c}_{h=1}, z_t \right], h^c_{t-1} \right) \tag{7}$$

where the hidden state $h^c_0$ is initialized as $\sum_i \mathtt{M}[i]$ and $\lambda$ is the parameters of $\mathtt{Ctrl}$. The GRU handles content-based input, empowered with pointer information to construct rich hidden state representations. Furthermore, the pointer-based data gives the GRU access to correct items in the sequence even when the memory content becomes different from the training due to encountering novel sequence length.

The Controller $\mathtt{Ctrl}$ uses the pointer values (mode 1), the related pointer values (mode 2) and the hidden state $h^c_t$ to make the final prediction. It simply concatenates them and forward to the $g^o(\cdot)$–a MLP, to generate the output token

$$\hat{y}^i_t \sim p_\Phi(y_t | X_i, z_t) = g^o_\lambda \left( \left[ \left\{ {}^* p^a_{t,h} \right\}^{H^a}_{h=1}, \left\{ {}^* p^c_{t,h} \right\}^{H^c}_{h=1}, h^c_t \right] \right)$$

The pointer values allow $g^o$ to fully utilize pointer information in producing the final output. $\Phi$ consists of the parameters of the $\mathtt{Encoder}_\theta$, Pointer Unit $\mathtt{PU}_\varphi$ and Controller $\mathtt{Ctrl}_\lambda$. Here, $\mathtt{Ctrl}$ can be put on top of

| Section | Task | Backbone |
|---------|------|----------|
| §3.1 | Algorithmic Reasoning | RNN (LSTM) |
| §3.2 | Dyck Language Recognition | MANN (SRNN) |
| §3.3 | Compositional Learning | Transformer (6 layers, RPE) |
| §3.4 | Other NLP Tasks (QA and Machine Translation) | Transformer (6 layers, BERT) |
| §3.5 | Algorithmic Reasoning and BIG-bench Tasks | LLM (Llama2-7B) |

Table 1: Task and backbone summary.

| Task | Key Metric | | Best Baseline | PANM (Ours) |
|------|-----------|---|---------------|-------------|
| Algorithmic Reasoning | ↑ Token Match (%) | 56.3 | (Other Max) | **68.2** |
| Dyck Language Recognition | ↑ Sequence Match (%) | 28.2 | (SRNN) | **70.0** |
| Compositional Learning | ↑ Sequence Match (%) | 82.6 | (U. TRM+RPE) | **88.6** |
| bAbI QA | ↑ Accuracy (%) | 77.5 | (TRM) | **83.0** |
| SQUAD QA | ↑ F1 (%) | 75.0 | (BERT) | **77.0** |
| Machine Translation | ↓ PPL | 42.7 | (TRM) | **32.7** |
| LLM Algorithmic Tasks | ↑ BLEU (%) | 14.7 | (Llama) | **62.7** |
| BIG-bench Tasks | ↑ Sequence Match (%) | 11.5 | (Llama) | **22.5** |

Table 2: Average performance summary with key metrics.

another decoder to process the decoding input $z_t = \texttt{Decoder}_\theta (y_{t-})$. For example, we can use Transformer as the Decoder (see Appendix C and D.3). The $\texttt{Encoder}_\theta$ and/or the $\texttt{Decoder}_\theta$ will be commonly referred to as the backbone model in which PANM is built upon. A summary of PANM's operation is given in Algo. 1 and Fig. 4 in the Appendix.

## 3   Experimental Results

In our experiments, we use two pointer variables in Mode-1 access and one for Mode-2 to balance between performance and computing cost ($H^a = 2$, $H^c = 1$, see more in Appendix C). The two Mode-1 pointer variables are initialized as the base and end addresses. All MLPs in PANM have 1 hidden layer of 128 dimensions. We use 256-dimensional GRUs for PU and Ctrl. The memory's address space has $b = 10$ bits, corresponding to a maximum of 1024 unique addresses, which is greater than any sequence length in the experiments.

In §3.1-3.3, our chosen tasks are representative of various types of symbolic reasoning and well-known benchmarks to measure the symbol-processing capability of machine learning models. *To showcase that these tasks are non-trivial, we report how Chat-GPT (Achiam et al., 2023) failed on our tasks in Appendix* D.7. To further validate this point, we finetune Llama2-7B (Touvron et al., 2023) on similar tasks and show that the LLM also fails to generalize even with finetuning on task data (see §3.5). In addition, we validate the contribution of PANM in other practical tasks in §3.4 and §3.5. We summarize the tasks and backbones used in our experiment in Table 1. We also summarize the final average performance of PNAM and the best baselines in Table 2.

**Baseline Choice** Despite some being simple, our baselines are still very strong methods in our studied tasks. For example, in our algorithmic reasoning, LSTM with attention or Pointer Networks are still dominant baselines, outperforming the more recent Transformers. There are also other sophisticated methods focusing generalization such as ESBN (Webb et al., 2020). In Dyck recognition, stack-based models are still SOTA because their inductive bias is suitable for the task. Experiments in §3.3 adopt (Universal) Transformer+Relative Positional Encoding (RPE), which is a very strong Transformer variant focusing on generalization. For experiments with LLMs, Llama2-7B stands out as the most capable open-source LLM that is suitable for our hardware. In our experiments, PANM-augmented models are ensured to have similar model size as the baselines and always share similar backbones for fair comparison.

| Task | Copy | Reverse | Mix | D. Recall | P. Sort | ID Sort |
|---|---|---|---|---|---|---|
| Other Max | 60.2 | 63.6 | 64.0 | 47.6 | 60.6 | 42.1 |
| PANM (Ours) | **74.8** | **73.6** | **81.2** | **52.8** | **67.8** | **59.2** |

Table 3: Algorithmic reasoning: mean sequence-level accuracy (%) over testing lengths Other Max is selected as the best numbers at each length mode from other baselines.

| Task | SCAN (L cut-off) | | | | | | | | | | | Math | |
|---|---|---|---|---|---|---|---|---|---|---|---|---|---|
| | 22 | 24 | 25 | 26 | 27 | 28 | 30 | 32 | 33 | 36 | 40 | a.s | p.v |
| U. TRM+RPE | 20 | 12 | 71 | **100** | **100** | **100** | **100** | **100** | **100** | **100** | **100** | **97** | 75 |
| TRM+RPE | 20 | 12 | 31 | 61 | **100** | **100** | **100** | 94 | **100** | **100** | **100** | 91 | 0 |
| U. TRM | 2 | 5 | 14 | 21 | 26 | 0 | 6 | 35 | 0 | 0 | 0 | 94 | 20 |
| TRM | 0 | 4 | 19 | 29 | 30 | 8 | 24 | 36 | 0 | 0 | 0 | 89 | 12 |
| PANM (Ours) | **22** | **47** | **100** | **100** | **100** | **100** | **100** | **100** | **100** | **100** | **100** | **97** | **86** |

Table 4: SCAN (Left): Exact match accuracy (%, median of 5 runs) on splits of various lengths. Mathematics (Right): mean accuracy over 5 runs. The baselines' numbers are from Csordás et al. (2021) and we run PANM using the authors' codebase.

## 3.1 Algorithmic Reasoning

In our first experiment, we study the class of symbol processing problems where an output sequence is generated by a predefined algorithm applied to any input sequence (e.g., copy and sort). The tokens in the sequences are symbols from 0 to 9. The input tokens can be coupled with meta information related to the task such as the priority score in Priority Sort task. During training, the input sequences have length up to $L$ tokens and can grow to $L+1$, $2(L+1)$, $4(L+1)$ or $8(L+1)$ during testing. Our setting is more challenging than previous generalization tests on algorithmic reasoning because of four reasons: (1) the task is 10-class classification, harder than binary prediction in Graves et al. (2014), (2) the testing data can be eight time longer than the training and the training length is limited to $L \approx 10$, which is harder than Grefenstette et al. (2015), (3) there is no curriculum learning as in Kurach et al. (2015), and (4) the training label is the one-hot value of the token, which can be confusing in case one token appears multiple times in the input sequence and tougher than using label as the index/location of the token as in Vinyals et al. (2015).

Here, we design several tasks. **Content-free tasks** involve permuting tokens in input sequence using certain position-based rules: First-In-First-Out (*Copy*), Last-In-First-Out (*Reverse*) and *Mix*. While the first two rules demand linear pointer manipulations (traverse the sequence from the head or tail, to output the target token), the last one uses a non-linear, length-dependent manipulation rule: if $t$ is odd, $y_t = x_{\lceil \frac{L}{2} \rceil}$; if $t$ is even, $y_t = x_1$. **Content-based tasks** need the input's token value together with symbol processing to arrange the output sequence. We introduce 3 tasks: *Dynamic Recall*, *Priority Sort* and *ID Sort*. Readers can find the details of these tasks in Appendix D.1.

**Baselines** are categorized into 4 groups: (1) Traditional RNNs such as LSTM (Hochreiter & Schmidhuber, 1997), (2) Sequential attention models: Content Attention (Bahdanau et al., 2014), Location Attention (Luong et al., 2015), Hybrid Attention (our baseline concatenates the attention vectors from content and location attention), (3) MANNs such as NTM (Graves et al., 2014), DNC (Graves et al., 2016), Neural Stack (Grefenstette et al., 2015) and Transformer (Vaswani et al., 2017), and (4) pointer-aware models: NRAM (Kurach et al., 2015), PtrNet (Vinyals et al., 2015), ESBN (Webb et al., 2020) and our method PANM. In this synthetic experiment, we adopt LSTM as the encoder for PANM. All baselines are trained with fixed number of steps (100K for ID Sort and 50K for the rest), which is enough for the training loss to converge. For each task, each baseline is trained 5 times with different random seeds and we use the best checkpoint on $L + 1$ mode validation to evaluate the baselines.

**Results** We report the average accuracy across different testing length for each task in Table 3. Overall, PANM significantly outperforms the best competitors ranging from 10-20% per task. Compared with individual baselines, the improvement is much higher (Appendix D.1). We illustrate how the pointer ma-

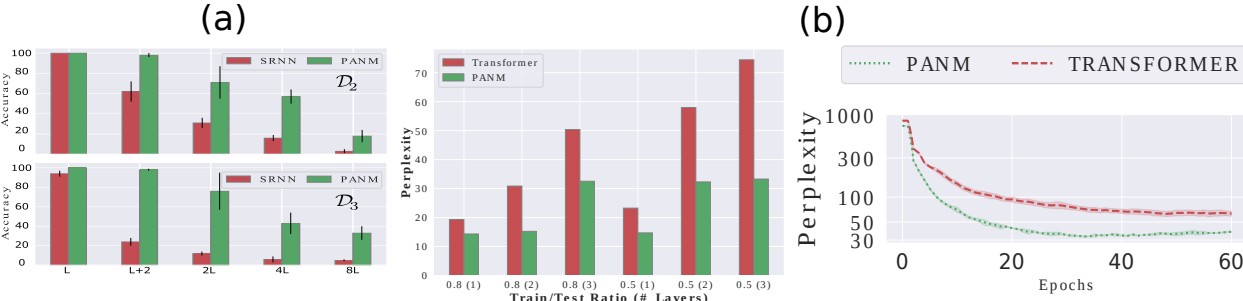

Figure 3: (a) Dyck: mean ± std. accuracy over 5 runs with different testing lengths. (b) Machine translation task: Perplexity on Multi30K dataset (the lower the better). We sort the sequences in the data by length and create 2 settings using train/test split of 0.8 and 0.5, respectively. The baselines are Transformer and PANM. **Left:** The best test perplexity over 2 settings for different number of Transformer's layers (1 to 3 layers). **Right:** an example of testing perplexity curves over training epochs for the case of 0.5 train/test split (2 layers) where we run 3 times and report the mean±std. The y-axis is visualized using log scale.

nipulation works for Copy and ID Sort in Fig. 2 (c) and (d). In Copy, only Mode-1 access is needed. As decoding step $t$ increases, Pointer Unit generates $p_t^a$ following the increment of the addresses as expected. In ID Sort, both Mode-1 and 2 are needed. The Pointer Unit generates $p_t^a$ incrementally to trace the input tokens from left to right (Mode 1). Then, the Mode-2 pointer $p_t^c$ is computed via attention to discover token with the same id, which will be the output at step $t$. Without Mode-2 access, PANM certainly fails this task. Experiments with varied number of heads are in Appendix D.6.

## 3.2 Dyck Language Recognition

Truly understanding the hidden law of context-free grammars such as Dyck ($\mathcal{D}_n$) is challenging for neural networks, even those with memory and attention (Yu et al., 2019). The language consists of strings with balanced pairs of brackets of different types ($|_1, |_2,...,|_n$), generated by the following rules: $S \rightarrow |_i S|_i$ with probability $p/n$ or $SS$ with probability $q$ or $\epsilon$ with probability $1 - p - q$. Here, $p, q$ are parameter of the language and $\epsilon$ is equivalent to EOS token. We follow the sequence prediction task and datasets in Suzgun et al. (2019) where the input is an unfinished Dyck string, and the output is the set of possible brackets for the next token, e.g., for $\mathcal{D}_2$, ([] → ( or ) or [. We follow the authors to enable set prediction by representing output $y_t$ as a multi-hot vector.

We adapt PANM to this autoregression task by masking M to ensure the decoder only see tokens up to the current decoding step. Since the token set is simple, we do not need to use any encoder, i.e., raw input tokens are stored in M. The SOTA baseline in this task is SRNN (Suzgun et al., 2019), an autoregressive model using stack as memory. We use this model as the decoder to make the setup of PANM close to SRNN. The only difference is that PANM has Pointer-Based Memory Access (Fig. 1 (b)). To make the task more challenging, we limit the maximum training length $L$ to 10 ($\mathcal{D}_2$) and 20 ($\mathcal{D}_3$) tokens, and the testing lengths are $L + 2$, $2L$, $4L$, $8L$. We choose $L$ as minimum numbers such that the model can perform decently on training data. The standard training and testing sizes are 5000. We train the models for 5 epochs and evaluate them on the training data at the end of each epoch to save model checkpoints. We use the best checkpoints for generalization test. Fig. 3 (a) reports the models' accuracy for $\mathcal{D}_2$ and $\mathcal{D}_3$. Under our extreme setting, SRNN generalization fades out quickly as test lengths increase, especially for $\mathcal{D}_3$ whereas PANM performance degradation happens at a much slower rate, outperforming SRNN by around 20% on average in both tasks at any test lengths.

## 3.3 Compositional Learning

**SCAN** In this task , one needs to map an input sentence into an output sequence of commands (Lake & Baroni, 2018). The sequences are compositional, consisting of reusable parts. For example, in one case, "jump twice" should be mapped to "JUMP JUMP" and in another, "walk twice" becomes "WALK WALK". We

| Model | Split | |
|---|---|---|
| | 0.8-0.2 | 0.5-0.5 |
| Transformer | $0.79 \pm 0.01$ | $0.76 \pm 0.01$ |
| U. TRM+ RPE | $0.80 \pm 0.02$ | $0.75 \pm 0.01$ |
| PANM (Ours) | $\mathbf{0.85} \pm \mathbf{0.02}$ | $\mathbf{0.81} \pm \mathbf{0.03}$ |

Table 5: bAbI QA: Testing accuracy (mean ± std.) over 5 runs.

focus on the "length split" datasets where the training sequences are shorter than the test ones with 11 length modes $L = 22, 24, .., 40$ (Newman et al., 2020). We adopt the benchmark, training procedure and baselines prepared by Csordás et al. (2021), which achieves strong results under standard s2s learning.

Here, our aim is not to break SOTA, which can be achieve by hybrid-symbolic architectures (Chen et al., 2020; Shaw et al., 2021). Instead, we focus on improving Transformer generalization in this task, hence the baselines are chosen as several variants of Transformers (TRM) targeted to sequence extrapolation, including those using Relative Positional Encoding (RPE (Dai et al., 2019)) and Universal Transformer (U. TRM (Dehghani et al., 2018)), which is an advanced Transformer variant that recurrently processes each token, and can dynamically adjust the number of processing steps. Following Csordás et al. (2021), each baseline is trained 5 times for 50K steps and the resulting model after training is used for evaluation (no validation). Here, we use Transformer as the `Encoder`, which is the same as the TRM, and stack the Controller to another Transform `Decoder` (see details in Appendix D.3). Hence, the only difference is the decoding where PANM leverages pointer manipulation.

Table 4 shows that PANM outperforms other baselines in the hardest settings when the training length is up-to 22, 24, and 25. For 22 and 24 cases, general models like PANM cannot show perfect generalization because some testing compositions is entirely discarded from the train set. In easier settings, PANM shares the perfect median accuracy with the sophisticated U. TRM + RPE although it does not use RPE. Remarkably, despite sharing the same encoder, TRM performs much worse than PANM and even fails to learn in easy modes (33, 36, 40), indicating the importance of pointer handling in this testbed. One problem for other baselines is the EOS decision (when to generate ending token), which requires length tracking (Newman et al., 2020). As they do not have content-free sequence iteration mechanisms, it is extremely hard to trace the length without overfitting to the training data. On the other hand, PANM can hypothetically generate pointer incrementally and capture the difference between the last and the first pointers, i.e. the input length, and infer the output sequence length based on that information.

**Mathematical Problems** We test our model on mathematics (Saxton et al., 2018) where the input/output are questions and answers about math and each token is a character. For example, `What is` $-5-110911? \rightarrow -110916$ (add_or_sub) and `What is the hundreds digit of 31253?` $\rightarrow 2$ (place_value). The task requires not only math reasoning, but also natural language understanding. We follow the training from Csordás et al. (2021) to conduct experiments on 2 subsets: `add_or_sub` (a.s) and `place_value` (p.v), and compare our method with Transformer-based baselines. Here, we focus on the extrapolating test set involving larger numbers, more numbers, more compositions, and thus longer input sequences than the training. We use TRM + RPE as the `Encoder` and the Controller is added to a normal TRM decoder. As shown in Table 4, on `place_value`, PANM does not suffer from performance crash as TRM + RPE (0% test accuracy, as admitted in the paper (Csordás et al., 2021) even though it uses the same encoder). PANM achieves similar results as U. TRM+ RPE on `add_or_sub` while outperforming it by 11% on `place_value`. We also report PANM +Transformer results in Appendix D.3.

### 3.4 Other NLP Tasks

**Question Answering** Our objective is to explore the PANM's generalization beyond obviously compositional data by applying it in a more practical setting of question answering. For this purpose, we utilize two datasets, namely bAbI (Weston et al., 2015) and SQUAD 1.1 (Rajpurkar et al., 2016) where the input sequence is a context paragraph and a question, and the output is the answer. To add complexity to the task, we ensure the length of test sequence is greater than that of the training by sorting the context paragraph by

| Model | Split | | | |
|---|---|---|---|---|
| | 0.8-0.2 | | 0.5-0.5 | |
| | F1 | EM | F1 | EM |
| BERT | 0.77 | 0.64 | 0.73 | 0.59 |
| PANM (Ours) | **0.78** | **0.65** | **0.76** | **0.61** |

Table 6: SQUAD 1.1: Testing accuracy after 3 epoch fine-tuning. F1 score and exact match (EM) follows the standard evaluation in Kenton & Toutanova (2019).

| Task | Model | Testing Length (# letters) | | | | | | |
|---|---|---|---|---|---|---|---|---|
| | | 10 | 20 | 40 | 100 | 200 | 500 | 1000 |
| Copy | Llama2-7B | $0\pm0$ | $0\pm0$ | $0\pm0$ | $0\pm0$ | $0\pm0$ | $0\pm0$ | $0\pm0$ |
| | Llama2-7B FT | $98\pm2$ | $16\pm4$ | $7\pm5$ | $0\pm0$ | $0\pm0$ | $0\pm0$ | $0\pm0$ |
| | PANM (Ours) | $\mathbf{99\pm1}$ | $\mathbf{99\pm1}$ | $\mathbf{97\pm1}$ | $\mathbf{86\pm2}$ | $\mathbf{85\pm2}$ | $\mathbf{87\pm2}$ | $\mathbf{92\pm2}$ |
| D. Recall | Llama2-7B | $0\pm0$ | $0\pm0$ | $0\pm0$ | $0\pm0$ | $0\pm0$ | $0\pm0$ | $0\pm0$ |
| | Llama2-7B FT | $86\pm4$ | $0\pm0$ | $0\pm0$ | $0\pm0$ | $0\pm0$ | $0\pm0$ | $0\pm0$ |
| | PANM (Ours) | $\mathbf{95\pm2}$ | $\mathbf{80\pm5}$ | $\mathbf{50\pm8}$ | $\mathbf{9\pm1}$ | $0\pm0$ | $0\pm0$ | $0\pm0$ |

Table 7: Synthetic Algorithmic Tasks: (zero-shot) LLM's mean $\pm$ std. BLEU accuracy (%) over 5 runs. Training length is 10 letters. Bold denotes best.

length and splitting the sorted data into 0.8/0.2 and 0.5/0.5 ratio. Details of the data/task are in Appendix D.4. In **bAbI**, we configure the PANM similarly to the one described in § 3.3 using Transformer backbone, and test the models after 100-epoch training. The models predict the answer tokens given the context and question tokens. As shown in Table 5 and Appendix Fig. 5 (right), PANM helps Transformer generalize better, consistently improving around 6% and 5% using 0.8/0.2 and 0.5/0.5 splits, respectively. Notably, PANM's testing loss is not diverged quickly as Transformer's, indicating PANM's capability of reducing overfitting. In **SQUAD**, we use BERT as the backbone to predict the start and the end of the answer as in Kenton & Toutanova (2019). PANM-assisted model outperforms the baselines by 1% and 2% exact match accuracy, respectively (Table 6). The improvement is significant as BERT is a big foundation model already pretrained with big data and robust against novel test data.

**Machine Translation** Here, we want to verify the PANM in machine translation and show that PANM can work with different number layers of Transformer. The results are presented in Fig. 3 (b) where we report the model perplexity on Multi30K (en-de) dataset. The 30K-sample dataset is sorted by input length and split into training and testing s.t. testing sequences are longer, similar to QA task. The results demonstrate PANM can consistently improve the generalization performance of Transformer across different split ratios and the number of encoder/decoder layers.

| Task | Model | Zero-shot | 5-shot |
|---|---|---|---|
| Arithmetic | Llama2-7B | $0\pm0$ | $32\pm2$ |
| | Llama2-7B FT | $17\pm4$ | $19\pm3$ |
| | PANM (Ours) | $\mathbf{33\pm2}$ | $\mathbf{38\pm2}$ |
| Abstract | Llama2-7B | $0\pm0$ | $0\pm0$ |
| | Llama2-7B FT | $0\pm0$ | $9\pm5$ |
| | PANM (Ours) | $\mathbf{3\pm1}$ | $\mathbf{16\pm4}$ |

Table 8: BIG-bench Tasks: LLM's mean $\pm$ std. Exact Match (%) over 5 runs. The finetuned models are trained and tested with shortest and longest sequences, respectively. Bold denotes best.

### 3.5 Scaled Generalization with LLMs

In this section, we explore the compatibility of PANM with LLMs. Specifically, we use Llama-2 (Touvron et al., 2023) as the backbone model and integrate the PANM layer on top of the final attention layer. There are 3 baselines: the pretrained LLM (Llama2-7B), the finetuned LLM (Llama2-7B FT) and the finetuned LLM+PANM (PANM). The models are finetuned with the same training configuration such as LoRA (Hu et al., 2021) and AdamW optimizer (Loshchilov & Hutter, 2018). The evaluation is executed using Language Model Evaluation Harness library (Gao et al., 2023).

**Synthetic Algorithmic Tasks** In the first test, we use Copy and Dynamic Recall (D. Recall) similar to those described in §3.1. We finetune both the LLM and LLM+PANM on a training dataset and then evaluate them on a separate testing dataset. In this test, each token is a letter sampled from the English alphabet (case sensitive), and the training data consists of 100,000 sequences, each of maximum 10 letters long, combined with an instruction introducing the task (see Appendix D.5). After fine-tuning, we evaluate the models on multiple testing sets, each of 1000 testing sequences with sequence lengths ranged from 10 to 1000 letters.

The results in Table 7 show that normal LLM finetuning generally outperforms the pretrained LLM in downstream tasks, but performance degrades rapidly as the testing length increases. Specifically, without PANM, performance drops to 0% at test lengths of 100 and 20 for Copy and Dynamic Recall, respectively. In contrast, PANM achieves 86% and 80% at these testing lengths. Notably, PANM maintains reasonable performance in the Copy task up to a test length of 1000. We also examine the evaluation setting when 5 few-shot examples are added to the prompt to enable LLM in-context learning. The results in Appendix Table 11 confirm that PANM significantly outperforms other baselines in this setting. Compared to the results of the same tasks in §3.1, we observe that PANM reaps more benefits in maintaining generalization with a stronger backbone like Llama2-7B, highlighting the versatility and potential of PANM.

**BIG-bench Tasks** Here, we focus on challenging benchmarks designed for LLMs. We pick the first two tasks from the BIG-bench benchmark (Srivastava et al., 2023), one is symbolic (Arithmetic) and one is not (Abstract). To ensure the experiment is both practical and challenging, we finetune the model using only a small subset of the training set containing the shortest input sequences. We then evaluate the model on a testing set with the longest input sequences to assess its generalization to longer sequences. Details of the chosen tasks are given in Appendix D.5.

The results in Table 8 demonstrate that PANM significantly improves the generalization accuracy of finetuned LLMs, outperforming the original finetuned LLM by a large margin of 16% and 3% in Arithmetic and Abstract, respectively. Even when augmented with 5 in-context examples, PANM maintains a substantial generalization gain (19% and 7%), confirming its effectiveness as a plug-and-play memory module for LLMs.

## 4 Related works

There are many attempts to augment neural networks with external memory (MANN) to improve their symbol-processing ability. Pioneers such as NTM (Graves et al., 2014) and DNC (Graves et al., 2016) propose computer-like memory read/write operations with content-based attention mechanisms, and thus in principle, can execute any symbolic rules. However, learning the hidden law end-to-end from sequence data is extremely difficult. Therefore, MANNs including Transformers (Vaswani et al., 2017), may fail miserably in out-of-distribution testbeds, especially length extrapolation (Delétang et al., 2022). Recent LLMs are good at reasoning and generalization, but bad at symbolic processing (Qian et al., 2023; Tang et al., 2023). We use LLMs only to show our task difficulty (Appendix D.7), not as a baseline, because they are not on the same scale as our method.

Many recent works advocate the use of specialized memory architectures such as stacks (Grefenstette et al., 2015; Hao et al., 2018; Suzgun et al., 2019), key-value memory (Webb et al., 2020; Le et al., 2020a) and improved attentions (Kurach et al., 2015; Russin et al., 2019; Dubois et al., 2020; Le et al., 2019). These methods employ different inductive biases in designing the memory and attention, *yet not following the two principles advocated by our paper*. Although they may work remarkably on certain synthetic tasks, they are

not examined on various benchmarks or compatible with different sequential backbones. Other orthogonal approaches focus on model initialization (Zhang et al., 2019), data augmentation (Andreas, 2020) or training details (Csordás et al., 2021). Besides differentiable models, there are major progress in compositional rule learning that leverage neuro-symbolic architectures (Nye et al., 2020; Shaw et al., 2021; Chen et al., 2020) or reinforcement learning (Liu et al., 2020). We have not compared our model with these task-specific methods, as our focus is on improving the systematic generalization of fundamental differentiable models.

Our approach is mainly related to key-value memory because the address bank can be viewed as the key and the data memory as the value. However, the key in other works is either learned through backpropagation (Le et al., 2020a; Le & Venkatesh, 2022) or computed based on the input data (Webb et al., 2020). In contrast, our "keys" are generated as fixed numbers (physical memory addresses– § 1's principle **I**), which is totally separated from the data and extendable to longer sequences. We argue that using addresses as keys is critical to symbol processing because it explicitly allows pointer assignment, dereference and arithmetic. A related generalization-enable scheme is to design positional encoding of tokens in a sequence (Vaswani et al., 2017; Dai et al., 2019; Li & McClelland, 2022). Recent advancements in this area have shown competitive results for length extrapolation up to 4 and 8 times the training length (Chen et al., 2024). Unlike these approaches, our method isolates the physical addresses (i.e., "positional encoding") from the data, allowing pointer transformation through time steps and absolutely detaching pointer manipulation from the input. Consequently, our method allows the freedom to point to any timestep, making it adaptable to a variety of backbones, not limited to Transformers.

## 5  Conclusion

We introduce a neural memory model called PANM, designed to manipulate pointers and learn symbol processing rules for improved length extrapolation. PANM separates symbols from data and utilizes an address bank to enable data-isolated pointer manipulation through address attention. PANM consistently outperforms strong baselines in tasks such as algorithm mining, compositional learning, mathematical reasoning, context-free grammar recognition, and practical NLP tasks, even when test sequences are significantly longer than training sequences. Remarkably, PANM achieves these results across various backbone models, including LSTMs, Transformers, and LLMs.

**Limitations**  While PANM provides significant benefits in enhancing sequence processing, it does come with certain limitations: (1) Computational Complexity: PANM introduces additional computational overhead, particularly when used with smaller models like LSTMs, where complexity can increase by 20-30%. For larger models, such as LLMs, this overhead is minimal, making PANM's operations relatively inexpensive. (2) Memory Requirements: The additional memory needed for the Address Bank and Pointer Unit is small relative to large backbone models. However, these components do contribute to overall memory usage, which may be a consideration in resource-constrained environments. (3) Implementation Challenges: Integrating PANM into existing architectures might require engineering effort. Ensuring seamless interaction between the backbone and PNAM and tuning PANM hyperparameters for optimal performance on specific tasks may require additional experimentation.

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

# Appendix

## A    More Discussion on Related Works

The proposed address attention in our paper is comparable to two known mechanisms: (1) location-based attention (Luong et al., 2015; Dubois et al., 2020) and (2) memory shifting (Graves et al., 2014; Yang, 2016). The former uses neural networks to produce attention weights to the memory/sequence, which cannot help when the memory grows during inference since the networks never learn to generate weights for the additional slots. Inspired by Turing Machine, the latter aims to shift the current attention weight associated with a memory slot to the next or previous slot. Shifting-like operations can handle any sequence length. However, it cannot simulate complicated manipulation rules. Unlike our PU design which obeys § 1's principle **II**, the attention weight and the network trained to shift it depend on the memory content M. That is detrimental to generalization since new content can disturb both the attention weight and the shifting network as the memory grows.

Another line of works tackles systematic generalization through meta-learning training (Lake, 2019), while our method employs standard supervised training. These approaches are complementary, with our method concentrating on enhancing model architecture rather than training procedures, making it applicable in diverse settings beyond SCAN tasks. Additionally, the study by Hu et al. (2020) addresses syntactic generalization (Hu et al., 2020), a different problem compared to our paper, which emphasizes length extrapolation across various benchmarks. Notably, our paper considers similar baselines, such as LSTM and Transformer, as those examined in the referenced papers. There are other lines of research targeting reasoning and generalization using image input(Wu et al., 2020; Eisermann et al., 2021). They are outside the scope of our paper, which specifically addresses generalization for longer sequences of text or discrete inputs

Our address bank and physical pointers can be viewed as some form of positional encoding. However, we do not use simple projections or embeddings to force the attention to be position-only. Instead, we aim to learn a series of transformations that simulate the position-based symbolic rules. At each time step, a new pointer ("position") is dynamically generated that reflects the manipulation rule required by the task (e.g. move to the next location), which is unlike the positional encoding approaches such as RPE (Dai et al., 2019) which aims to provide the model with information on the relative position or distance of the timesteps. We summarise the difference between our method and Transformer in Table 9.

## B    More Discussion on Base Address Sampling Mechanism

We provide a simple example to illustrate how base address sampling help in generalization. Assume the training sequence length is 10, and the desired manipulation is $p' = p + 1$ (copy task). Assume the possible address range is $0, 1, ..., 19$, which is bigger than any sequence length. If $a_B = 0$, the training address bank contains addresses: $0, 1, ...8, 9$. Without base address sampling, the model always sees the training address bank of $0, 1, ...8, 9$ and thus can only learn manipulating function for $0 \leq p \leq 9$, thereby failing when testing address bank includes addresses larger than 9.

Thanks to base address sampling, at some point of training, $a_B = 10$, the training address bank is $10, 11, ...13, 19$. The manipulating function sees $p > 9$ and can learn to transform $p' = p + 1$ for $p > 9$, e.g., transform $p = 10 \rightarrow p' = 11$. The learning happens because the pointer's value ($^*p$) is used to predict the output sequence. The task loss will reward $^*p$ that follows the rule, and update the Pointer Unit such that it transforms the $p$ following the rule to minimize the loss. During testing, the input length can be 12, we set $a_B = 0$ and the address bank is $0, 1, ...., 10, 11$. The learned manipulation can still be applied to new locations 10th, and 11th.

We can prove that the complexity of exposing all addresses to the model is practically small compared to the normal training. Assume the training input sequence length is $L$, and the number of possible addresses is $L_{max}$. Here, $L_{max}$ indicates the possible testing length that the model can handle. When $L_{max} \rightarrow \infty$, the expected number of samples required for exposing all addresses is $O(n \log n)$ where $n = L_{max}/L$ (we can formulate this problem as Coupon collector's problem). For example, in even an extreme address range of $L_{max} = 10^6$ (in practice we rarely need that big range) to train input sequences of length 10, we only need

| Difference | Transformer | PANM (Our) |
|---|---|---|
| Key Generation | Keys are computed based on input data. Hence, when meeting novel data during testing, Transformer will observe novel keys, and cannot work properly. | The keys in our approach are generated as fixed numbers, specifically physical memory addresses. These keys are entirely separate from the data. |
| Extendable to Longer Sequences | The dimension of attention weights varies with input length, making arithmetic transformations on these attention weights infeasible as the sequence length increases. | The fixed nature of our physical addresses allows our pointers to be easily manipulated and extendable to longer sequences. |
| Symbol Processing Advantages | The use of attention weights as implicit pointers may lack the explicitness needed for effective symbol processing. | Using physical addresses as keys in our approach is crucial for symbol processing as it explicitly allows pointer assignment, dereference, and arithmetic operations. |
| Physical Address vs Positional Encoding | Positional encoding can be generated independently from data. However, they are not separated from the input data as our physical addresses. There is no explicit mechanism in Transformer to attend only to these positional encodings or to transform pointers to point to these positional encodings from one step to another. | Our physical addresses are detached from the data, supporting the transformation of pointers through timesteps and isolating pointer manipulation from the input. |

Table 9: PANM vs Transformer

to sample $10^5 \log 10^5$ sequences, which is often smaller than the size of the training datasets. Empirically, in our experiments, we always train our method with the same number of batch size and training steps as other baselines to ensure a fair comparison, and we realize that it is always possible to expose all the addresses to our model during training.

## C  More Discussion on Model and Architecture

We can see that it is critical to have $H^a \geq 2$ and $a_B, a_E \in \left\{p_{0,h}^a\right\}_{h=1}^{H^a}$ to achieve generalization using pointer arithmetic. In other words, if $a_B, a_E \notin \left\{p_{0,h}^a\right\}_{h=1}^{H^a}$, we can always find a task to make PANM fail to generalize. For example, if $a_E \notin \left\{p_{0,h}^a\right\}_{h=1}^{H^a}$, PANM cannot generalize in Reverse task. To see that, without loss of generality, we assume PANM only learn to produce the last token at address $a_E$ using information from some initial addresses $p' \in \left\{p_{0,h}^a\right\}_{h=1}^{H^a}$ such that $p' \neq a_E$. During training, the learned pointer arithmetic to perform Reverse at the first step of the decoding to produce value $y_1$ can only be a function of $p'$: $y_1 = {}^*a_E = {}^* f(p')$, that is, $a_E = f(p')$. During testing, $a_E$ can receive arbitrary value, so for whatever learned $f$, we can always find a test sequence such that $a_E \neq f(p') \; \forall f$ because $a_E \neq p'$. A similar argument can be used for $a_B$ and Copy task.

In the main manuscript, we only experiment with 1 Mode-2 pointer ($H^c = 1$). If $H^c = 0$, obviously PANM will fail in tasks such as ID Sort. Using more $H^c$ and $H^a$ can still be beneficial in exchange for slower computation (see Appendix D.6). In all experiments, we use 256-dimensional GRUs for the PU and Ctrl. The encoder and decoder (to stack the Controller on) can vary across tasks. The general plug-and-play framework is illustrated in Fig. 4. We also summarize operations of our model in Algo. 1.

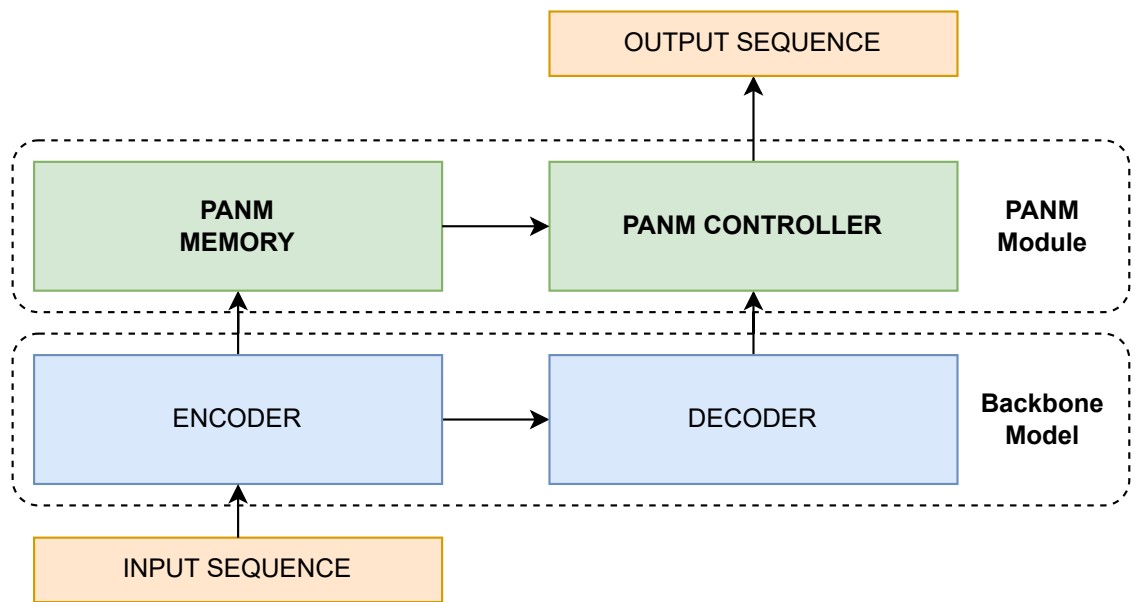

Figure 4: PANM as a plug-and-play architecture. The encoder and decoder can be any model (LSTM, Transformer or BERT). PANM Controller can be used as the last layer of the Decoder to access the memory during decoding. To reduce the number of parameters of the augmented architecture, the decoder's number of layers can be decreased.

## D    Experimental Details

All datasets and public codebases used are licensed under Apache or MIT License. We trained all the models on a single Tesla V100-SXM2 GPU. The running time of PANM depends on the `Encoder` and the specific tasks. Overall, with 2 Mode-1 pointers and 1 Mode-2 pointer, PANM operates at 70-80% of the speed of the backbone model. For instance, in the Copy task, PANM achieves 15 iterations per second compared to LSTM's 20 iterations per second. When using a Transformer Encoder, PANM runs at 77 iterations per second, whereas the Transformer runs at 90 iterations per second. Notably, the larger the backbone, the smaller the difference in running time. In LLM experiments, there is no significant difference in training/inference speed between settings with PANM and without PANM, with speeds of 15.2 iterations per second and 15.6 iterations per second, respectively.

### D.1    Algorithmic Reasoning

We first give the details of the content-based tasks below.

In **Dynamic Recall**, an arbitrary input token is chosen as the query and is added to the end of the input sequence. Depending on the length of the input, a.k.a, odd or even, the first target token will be on the left or right of the query, following its succeeding tokens in the input. This task requires both content matching (find query token in the input sequence) and position-based access (shift left or right).

In **Priority Sort**, each input token is associated with a priority score sampled from the standard normal distribution. The target output will be tokens from the input sequence sorted ascending by their the score. This task can be solved in many ways and likely needs complicated symbol processing such as looping through items in the sequence and comparing the score of tokens.

Finally, in **ID Sort**, each input token is augmented with an id feature vector sampled from standard multivariate normal distribution such that every 2 tokens share one id. For example, with input $x_1, x_2, x_3, x_4$, $x_1$ and $x_4$ may share one id while $x_2$ and $x_3$ shares another id. The pairing is chosen randomly. The output token at position $i$-th will be the input token that share id with the $i$-th input token. The correct output

for the earlier example is $x_4, x_3, x_2, x_1$. This task is specifically designed to test the ability to learn Mode 2 pointer-based memory access.

In this task, we implement the baselines such as LSTM, attention models and Transformer using Pytorch library. The hidden state dimension for these models are set to 512, which results in around 1-3 millions parameters. We tuned the number of layers of the encoder/decoder for these baselines in Copy task, and realized that 1-layer gave the best performance. For NTM and DNC, we use public repositories[1] with default controller's hidden state of 256 dimensions and 128-slot external memory, which results in around 1.2 millions parameters. We use the ESBN's author codebase [2] with default parameter setting, resulting in $\approx$1.2 million parameters. For PtrNet, since we do not use token index as the training label, we produce the predicted token by performing weighted sum the input tokens using the PtrNet's attention weights. PtrNet's hyperparameters are the same as attention models. We could not find the authors' code for Neural Stack and NRAM so we implemented them and tuned hyperparameters for the Copy task at length $L$ such that the model sizes are about 1.1 million parameters. In this task PANM uses LSTM with hidden state of 256 as the `Encoder` and does not stack the Controller on any decoder models, resulting in $\approx$1.1 million parameters.

In this experiment, all the models are trained without teacher forcing as in Graves et al. (2014), i.e, the input to the decoder is zero ($z_t = 0$). The detailed average accuracy (mean $\pm$ std.) of each method together with the actual length of each testing mode are reported in Tables 13-18.

Overall, PANM observes significant improvement ranging from 10-20% on each task. We note that when compared with individual baselines, the improvement is much higher. Consider Copy as an example (Fig. 2a), PANM outperforms the worst baseline Transformer by around 60% at $2(L+1)$ and 30% at $4(L+1)$, respectively. As stated earlier that our tasks are challenging, thus, originally strong baselines such as NTM, DNC, and Neural Stack do not generalize well at extreme lengths, especially in ID Sort. ID Sort is trickier than content-free tasks, making some baselines fail at length $L$ even though it is in the training data. The best other model in this case is Content Attention, which clearly underperforms our PANM from few % to 50% (Fig. 2b). Without curriculum learning and under the 10-class prediction setting, methods that use implicit pointers, such as PtrNet, NRAM, and ESBN, demonstrate mediocre performance on average when compared to PANM. Furthermore, PANM also outperforms in length-dependent tasks (Mix, D. Recall), indicating that it can track the sequence length in extrapolation. We hypothesize that PANM's content–free pointer generation mechanism to simulate list iteration makes it possible.

In Copy, only Mode-1 access is needed. As decoding step $t$ increases, Pointer Unit generates $p_t^a$ following the increment of the addresses as expected. That said, for several steps, the address attention is not sharp, showing other addresses pointed by the pointer, which is not surprising since we use soft attention and it is hard for a neural network to learn the exact rule: $p_{t+1}^a = p_t^a + 1$. This problem gets worse as test length increases as the error accumulates, especially when the same token can appear many times, which confuses the model. This explains why PANM's performance drops clearly in the hardest case $8(L+1)$. Yet, it is still significantly better than others whose results are near random prediction.

## D.2 Dyck Language Recognition

In this task, we adopt the SRNN code from Suzgun et al. (2019)[3] using the default parameters. As explained in the main text, this task is auto-regression, hence, $z_t = \hat{y}_{t-1}$. PANM adopts SRNN (an auto-regressive model) as the encoder and does not stack the Controller on any decoder models. The result is visualized in Fig. 5 (left).

## D.3 Conpositional Learning

In this task, we adopt the code from Csordás et al. (2021)[4] using the default parameters. When using Transformer Encoder, we need to have Transformer-like decoder to align the token representation of the encoding and decoding phases. As such, in SCAN, we utilize the 3-layer Transformer decoder, replace

---

[1]`https://github.com/thaihungle/SAM`
[2]`https://github.com/taylorwwebb/emergent_symbols`
[3]`https://github.com/suzgunmirac/marnns`
[4]`https://github.com/RobertCsordas/transformer_generalization`

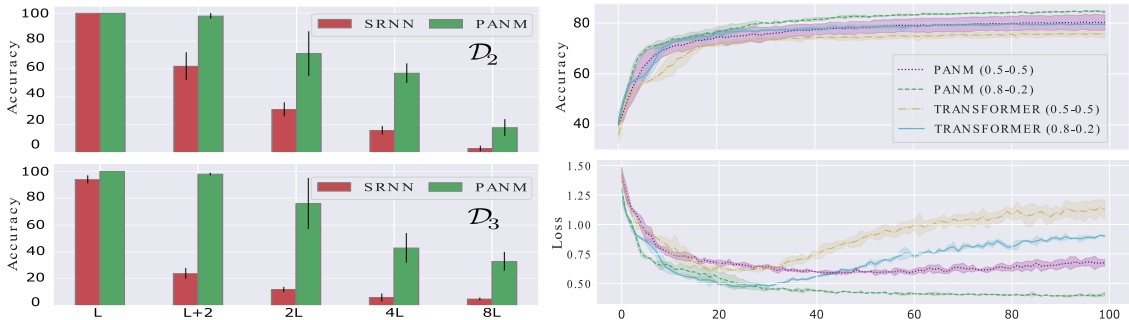

Figure 5: Dyck (Left): mean ± std. accuracy over 5 runs with different testing lengths. bAbI QA (Right): mean ± std. testing accuracy and cross-entropy loss across 100 training epochs over 5 runs.

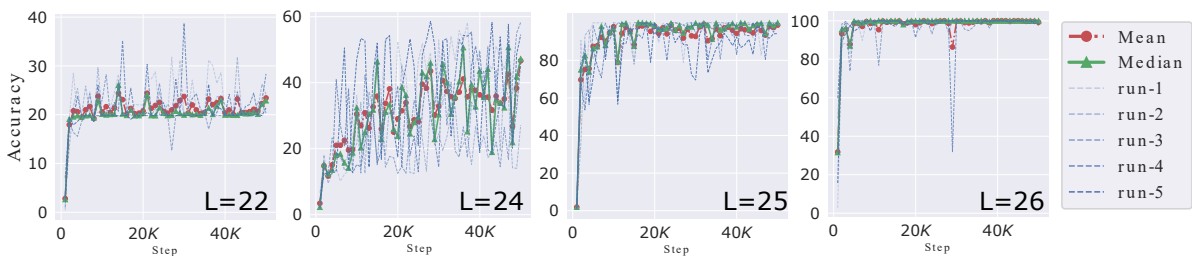

Figure 6: SCAN: PANM's exemplar learning curves.

its last layer by the Controller. Formally, $z_t$ in Eq. 7 becomes Decoder($y_{t-}$) where the Decoder is a 2-layer Transformer. In Mathematics reasoning task, we use similar integration except that the Encoder is Transformer with relative positional encoding (TRM + RPE). By reducing the number of decoding layer, we ensure PANM's hyperparameter number equivalent to that of the Transformer baseline (12M). All models are trained with teacher forcing as in Csordás et al. (2021).

**SCAN** The training size is 16990 and the test size is 3920. SCAN is a well-known and standard benchmark for testing compositional learning and generalization in sequential models. One property of this dataset is that a new length often contains new rules that must be captured by the model to ensure generalization, and thus, if the model fails to learn a hidden rule, its performance may drop significantly from one length split to another. Fig. 6 illustrates PANM's testing accuracy curves when $L = 22, 24, 25, 26$. Other learning curves for $L > 26$ looks similar to $L = 26$ where PANM easily solves the task perfectly.

**Mathematical Problems** Table 19 reports the accuracy with mean and standard deviation. Here, we augment TRM and TRM+RPE with PANM. Both shows improvement, especially for TRM+RPE, indicating that PANM is compatible with other methods designed to improve generalization in Transformer.

### D.4 Other NLP Tasks

The bAbI dataset consists of 20 synthetic tasks that evaluate various reasoning skills. To prepare the data for each task, we combine train/valid/test into a single set and sort it by length and split it into training and testing sets, as described in the main text. We train the models jointly on all 20 tasks and measure the accuracy of their answers, which are considered correct only if they match the ground truth answer perfectly. The training/evaluation follows exactly the standard protocol presented in (Le et al., 2020b). The Transformer used here has 8 heads, 3 layers of encoding, 3 layers of decoding, and hidden dimensions of 512. PANM uses the same Transformer backbone except that the decoder has 2 layers to make the model size equivalent. We run each model 5 times to report the mean and standard deviation as in Fig. 5 (right). Table 5 reports the detailed numbers.

The SQUAD dataset contains more than 100K realistic context/question-answer pairs. Again, we combine train/test into a single set and sort it by length and split into new train/test sets. Following Kenton & Toutanova (2019), we use BERT model (`https://huggingface.co/bert-base-uncased`) to predict the start and end location of the answer, and finetune the model with the same setting (e.g., 3 epochs with a learning rate of 5e-5) except that our batch size is 16 to fit with our GPU. PANM appends the Controller to BERT to predict the start and end. Both BERT and PANM have around 114 million parameters. Table 6 reports the detailed numbers.

### D.5 LLM Tasks

The training follows torchtune library: `https://github.com/pytorch/torchtune` using the default training hyperparameters. For experiments using LoRA finetuning, we use LoRA with the following configuration:

- Rank: 8

- $\alpha$: 8

- Target layers: q_proj,v_proj

The optimizer configuration is:

- Optimizer:
  - AdamW with weight_decay of 0.01
  - Learning rate: 3e-4

- Learning rate scheduler:
  - Cosine scheduling with 100 warm up steps

Training configuration:

- The batch size is 8 with 4 gradient accumulation steps

- Number of epochs: 3

Below is the example of the prompt given to the LLM in the used tasks.

```
Copy
Prompt:  "This is copy task.  Given this:  F b P i P B m s G j \n Your answer:"
Target:  "F b P i P B m s G j"
```

```
Dynamic Recall
Prompt:  "This is dynamic recall task.  You are given a sequence of characters.  The
last character is the query and you need to retrieve the first character after or
before the query in the sequence if the number of the characters in the sequence is
odd or even, respectively.  Given this:  E V y D m I e V u f _ V \n Your answer:"
Target:  "E"
```

```
Bigbench Arithmetic
Prompt:  "Q: What is 63539 times 88329?\n choice:  1572879084\n choice:  2152788438\n
choice:  banana\n choice:  34615225\n choice:  5612336331\n choice:  556451339991\n
choice:  house\nA:"
Target:  "5612336331"
```

*Bigbench Abstract*
Prompt:  "Given the choice:  People who live in glass houses shouldn't throw stones\nGood things come in small packages\nDon't put new wine into old bottles\nBeat swords into ploughshares\nPractice what you preach\nA man who is his own lawyer has a fool for his client\nA barking dog never bites\nHe who pays the piper calls the tune\nApril showers bring forth May flowers\nFish always stink from the head down\nAbsolute power corrupts absolutely\nJack of all trades, master of none\nThe wages of sin is death\nLove of money is the root of all evil\nChristmas comes but once a year\nWhat's sauce for the goose is sauce for the gander\nIt's the squeaky wheel that gets the grease\nPractice makes perfect\nThe age of miracles is past\nAn army marches on its stomach\nIf the mountain won't come to Mohammed, then Mohammed must go to the mountain\nA soft answer turneth away wrath\nHoney catches more flies than vinegar\nDead men tell no tales\nBuild a better mousetrap and the world will beat a path to your doorLink to proverb\nRevenge is a dish best served cold\nAll publicity is good publicity\nDon't meet troubles half-way\nFirst impressions are the most lasting\nIt takes two to tango\nAn Englishman's home is his castle\nLittle things please little minds\nCut your coat to suit your cloth\nLook before you leap\nCheaters never win and winners never cheat\nA golden key can open any door\nA prophet is not recognized in his own land\nSilence is golden\nSuccess has many fathers, while failure is an orphan\nGod helps those who help themselves\nLaughter is the best medicine\nThere's no accounting for tastes\nDo unto others as you would have them do to you\nA stitch in time saves nine\nNever judge a book by its cover\nHard cases make bad law\nA house divided against itself cannot stand\nTime is money\nThat which does not kill us makes us stronger\nSeek and you shall find\nFailing to plan is planning to fail\nThe cobbler always wears the worst shoes\nYou are never too old to learn\nHe who laughs last laughs longest\nDon't shoot the messenger\nGood things come to those that wait\nSeeing is believing\nHindsight is always twenty-twenty\nOnly fools and horses work\nFlattery will get you nowhere\nNothing new under the sun\nWhat can't be cured must be endured\nA cat may look at a king\nNo rest for the wicked\nVirtue is its own reward\nOnce bitten, twice shy\nHaste makes waste\nA nod's as good as a wink to a blind horse\nThe best things in life are free\nStrike while the iron is hot\nFrom the sublime to the ridiculous is only one step\nDistance lends enchantment to the view\nTo err is human; to forgive divine\nHe who hesitates is lost\nYou can't hold with the hare and run with the hounds\nA poor workman always blames his tools\nFeed a cold and starve a fever\nFinders keepers, losers weepers\nMake haste slowly\nGreat oaks from little acorns grow\nLive for today for tomorrow never comes\nIt takes a thief to catch a thief\nIt's better to light a candle than to curse the darkness\nThere's honour among thieves\nMoney doesn't grow on trees\nBetween two stools one falls to the ground\nGive credit where credit is due\nThe apple never falls far from the tree\nNothing is certain but death and taxes\nCleanliness is next to godliness\nBad news travels fast\nLife is what you make it\nThe customer is always right\nOne hand washes the other\nDon't let the grass grow under your feet\nThe end justifies the means\nFor want of a nail the shoe was lost; for want of a shoe the horse was lost; and for want of a horse the man was lost\nPossession is nine points of the law\nAn apple a day keeps the doctor away\nMarriages are made in heaven\n.
In what follows, we provide short narratives, each of which illustrates a common proverb.  \nNarrative:  Today was Kim's birthday, and she wanted to celebrate with all of her friends.  Cindy told her she couldn't join them, because she wanted to get some work done.  Kim asked if she could please take some time away from work to have fun and celebrate, but Cindy wouldn't.  She just talked about all of her future plans that she had to prepare for.  Later that night, Cindy saw all of the pictures from Kim's birthday celebration, and she felt sad.  None of her future plans were happening any time soon, and in the meantime, she missed the party.\nThis narrative is a good illustration of the following proverb:"
Target:  "Live for today for tomorrow never comes"

| Task | Train/Test Size | Train Max Length | Test Max Length |
|------|-----------------|------------------|-----------------|
| bigbench_arithmetic generate_until | 1000/1000 | 126 characters | 178 characters |
| understanding_bigbench_abstract narrative_generate_until | 700/200 | 828 characters | 1507 characters |

Table 10: BIG-bench tasks.

| Task | Model | Testing Length (# letters) | | | | | | |
|------|-------|----|----|----|-----|-----|-----|---------|
| | | 10 | 20 | 40 | 100 | 200 | 500 | $1000^{(*)}$ |
| Copy | Llama2-7B | $100{\pm}0$ | $99{\pm}1$ | $98{\pm}1$ | $98{\pm}1$ | $95{\pm}2$ | $8{\pm}1$ | $0{\pm}0$ |
| | Llama2-7B FT | $100{\pm}0$ | $99{\pm}1$ | $99{\pm}1$ | $98{\pm}1$ | $97{\pm}2$ | $31{\pm}3$ | $0{\pm}0$ |
| | PANM (Ours) | $100{\pm}0$ | $\mathbf{100{\pm}0}$ | $\mathbf{100{\pm}0}$ | $\mathbf{100{\pm}0}$ | $\mathbf{99{\pm}1}$ | $\mathbf{39{\pm}3}$ | $\mathbf{3{\pm}0}$ |
| D. Recall | Llama2-7B | $10{\pm}1$ | $6{\pm}1$ | $1{\pm}1$ | $3{\pm}1$ | $5{\pm}1$ | $3{\pm}1$ | $1{\pm}1$ |
| | Llama2-7B FT | $12{\pm}2$ | $4{\pm}1$ | $0{\pm}0$ | $3{\pm}1$ | $5{\pm}2$ | $2{\pm}2$ | $1{\pm}1$ |
| | PANM (Ours) | $\mathbf{60{\pm}7}$ | $\mathbf{38{\pm}5}$ | $\mathbf{18{\pm}3}$ | $\mathbf{9{\pm}1}$ | $\mathbf{11{\pm}1}$ | $\mathbf{5{\pm}1}$ | $\mathbf{3{\pm}1}$ |

Table 11: Synthetic Algorithmic Tasks: (5-shot) LLM's mean $\pm$ std. BLEU accuracy(%) over 5 runs. Training length is 10. ($*$) In this case, input overflow happens because maximum input length for Llama2 is 4096 tokens.

## D.6 Additional Experiments

**Generalization to Shorter Sequence** Here, we explore whether PANM can maintain its performance in a long-to-short generalization scenario. We selected a challenging synthetic task, Mix, and trained both PANM and Location Attention (the best baseline for this task) on sequences of length 20-30, testing them on sequences of length 10. The training and testing curves in Fig. 7 (left) reveal that while both models achieve near-perfect training results, only PANM shows smooth learning and signs of generalization to the test length of 10, with an improvement margin of nearly 14% compared to Location Attention.

Furthermore, in a more practical Machine Translation task, we trained both PANM and Transformer (the best baseline for this task) on 50% longer sequences and tested on the remaining shorter sequences using the same 2-layer Transformer backbone. As shown in Fig.7 (right), the Transformer model overfits and fails to generalize to shorter sequences. In contrast, PANM achieves reasonable perplexity on the test data, outperforming the best Transformer result by approximately 25 points.

**Pointer Hyperparameters** In this section, we confirm the logic presented in Appendix C by performing experiments that involve varying the number and type of pointers.

**Mode-1 Pointers** We test the PANM version in § 3.1 with $H^a = 0, 1, 2, 3$ on Copy, Reverse. We do not use Mode-2 pointer here to avoid confusion ($H^c = 0$). Fig. 8 plots the testing accuracy over training time. As $H^a = 0$, there is no pointer information for the Controller, PANM should be equivalent to an GRU and fail to generalize. As $H^a = 1$, the only pointer is initialized either with the base or end address. As shown in Fig., PANM cannot generalize in both Copy and Reverse tasks with single Mode-1 pointer, which is proved in Appendix C. In the case $H^a = 3$, we initialize them with the base, end and middle addresses. We observe that increasing $H^a$ to 3 slightly reduces the performance in these tasks. We speculate that too many Mode-1 pointers make the learning harder; in particular, learning to manipulate the third pointer may interfere with that of the first or second pointer, which are more important in these tasks. Generally, most tasks only require list iterations from the head to the tail or vice versa. Hence, we keep $H^a = 2$ in all experiments to save the computation cost.

**Mode-2 Pointers** We fix $H^a = 2$, and vary $H^c = 0, 1, 2$ on Copy, Priority Sort, ID Sort. As shown in Fig. 9, without Mode-2 pointers ($H^c = 0$), generalization in Priority Sort and ID Sort is reduced significantly by 50% and 30%, respectively because these tasks focus more on the content of the input sequence and often demand comparing the content of different tokens. Interestingly, a content-free task like Copy also suffers from performance drop if there is no Mode-2 pointer. Specifically, we find out that for 2/5 runs,

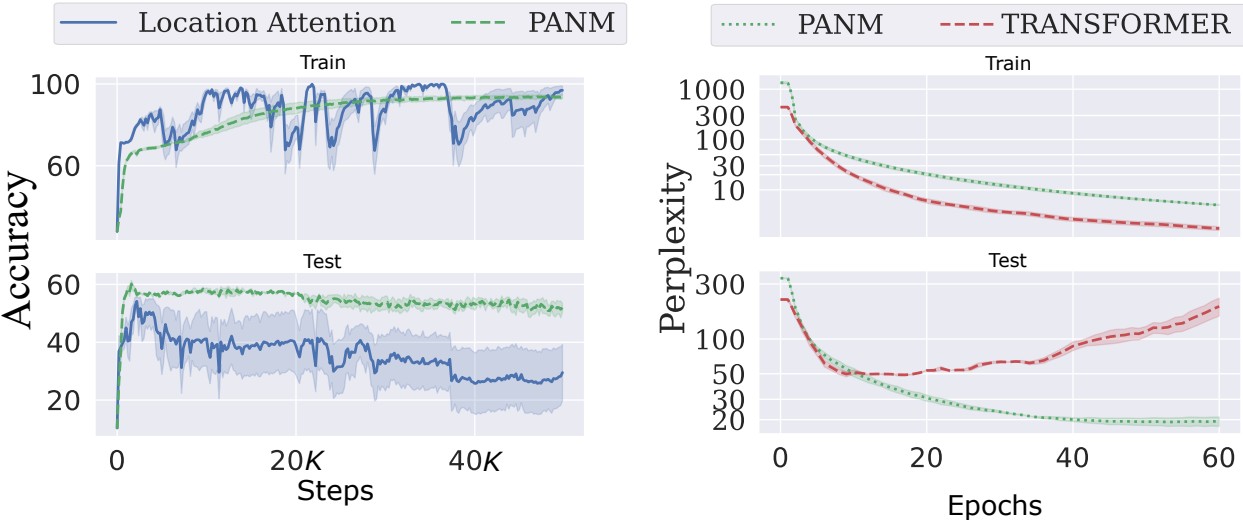

Figure 7: Long-to-short generalization: Mix (left) and Machine Translation (right) learning curves. The curves are mean ± std. over 5 runs.

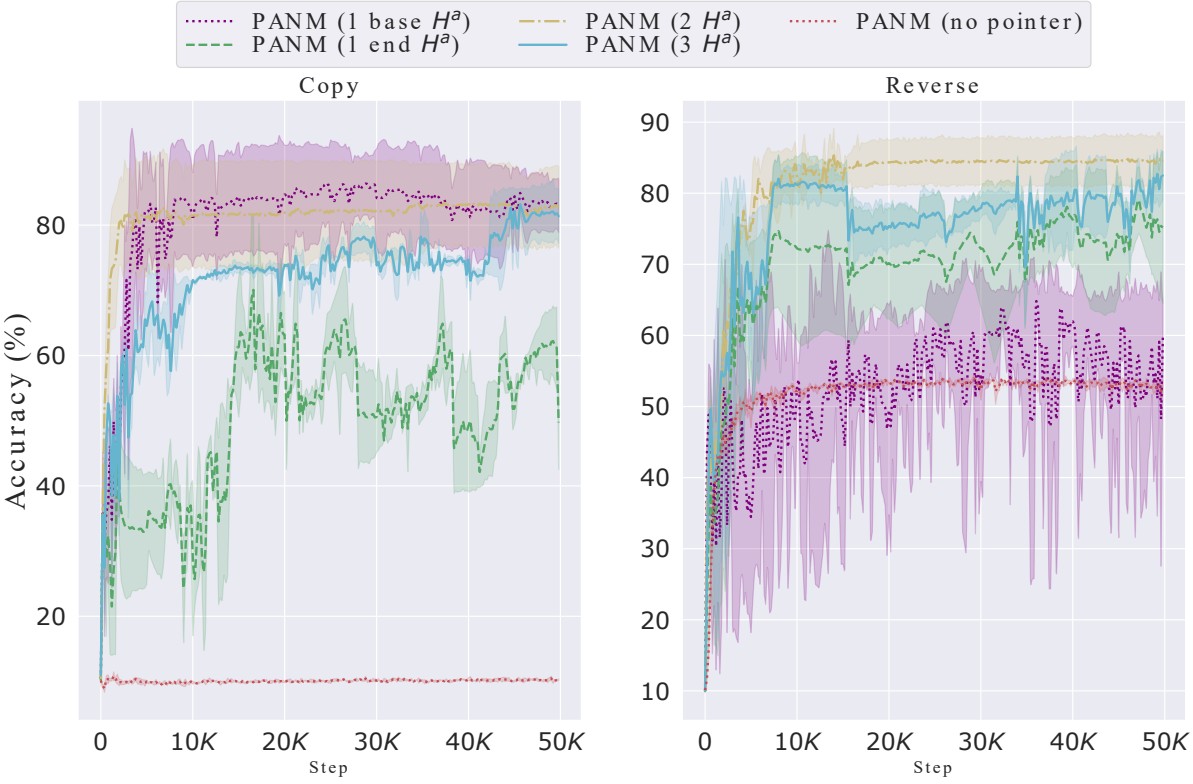

Figure 8: Testing accuracy (mean ± std.) at 2(L+1) length over training steps. Different configurations of Mode-1 pointers are trained and evaluated 5 times.

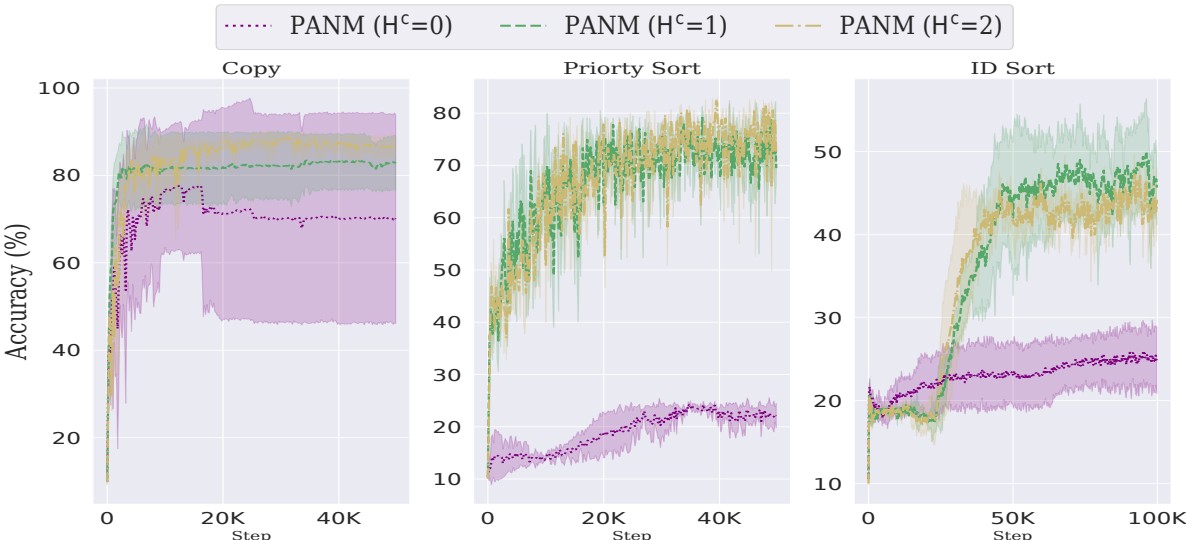

Figure 9: Testing accuracy (mean ± std.) at 2(L+1) length over training steps. Different configurations of Mode-2 pointers are trained and evaluated 5 times.

the model converges to a suboptimal solution, leading to high variance and slightly lower mean accuracy. Perhaps, Mode-2 pointer allows the decoder to access the input instantly (like content-based attention), avoid forgetting, and thus, improve the prediction as the sequence is longer. Having more Mode-2 pointers generally improves the generalization in Copy and Priority Sort, yet the gain is small for $H^c = 2$, or even negative in ID Sort. Therefore, we trade-off performance with computing efficiency by setting $H^c = 1$ in our experiments.

### D.7   Failures of Chat-GPT in Our Tasks

Large Language Models (LLMs), especially Chat-GPT, have shown remarkable results in reasoning and generalization. Directly comparing Chat-GPT with other models used in our experiments would be unfair because Chat-GPT was not directly trained with our datasets and it has much more parameters than our model. Therefore, in this section, we merely use Chat-GPT as a tool to verify that our chosen tasks, despite being simple, are non-trivial. The evaluated tasks are algorithmic reasoning and SCAN. We do not examine Dyck recognition because the output encoding is complicated to represent in text. Other datasets are more common and likely to be used for training Chat-GPT, thus, are not suitable for generalization test. For example, in Mathematics task, if we ask Chat-GPT the question from the data `What is the hundreds digit of 31253`?, it provide the correct answer (2). However, slightly modifying the question to ensure it does not appear in the training and testing set will successfully fool Chat-GPT:

- Example 1:
  - Prompt: What is the hundreds digit of 312537?
  - Chat-GPT answer: The hundreds digit of the number 312537 is 2.

- Example 2:
  - Prompt: What is the hundreds digit of 319253?
  - Chat-GPT answer: The hundreds digit of the number 319253 is 9.

We use Open AI's Chat-GPT 3.5 version September 2023 and evaluate the model on our data using few-shot example prompts, following the format:

Examples:
input $x_1^1, x_2^1, ...$ output $y_1^1, y_2^1, ...$

| Task | Chat-GPT | Failure example | | | PANM |
| --- | --- | --- | --- | --- | --- |
| | | Input | Chat-GPT Output | True Output | |
| Copy | 100% | | N/A | | 84% |
| Reverse | 69% | $%&&$%^@%# | %#^@^%$&&%$ | #%@^%$&&%$ | 84% |
| Mix | 42% | $%&&$%^@%# | %#^&&$%$@%& | $%$&$%$@$# | 98% |
| Dynamic Recall | 14% | $(&&$#^@%# % | $ | @ | 45% |

Table 12: Failure of Chat-GPT on algorithmic reasoning test cases of length $2L$. Token-level accuracy is reported. We do not test Chat-GPT on Priority and ID sort because they have complicated token representations. PANM results cannot be directly compared, and shown for reference only.

input $x_1^2, x_2^2, ...$ output $y_1^2, y_2^2, ...$
...
Question:
input $x_1, x_2, ...$ output

**Algorithmic Reasoning**  To ensure that Chat-GPT does not memorize the output answer from its vast training data, we use non-digit symbols: ~!@#$%^&*( as 10 tokens of the datasets. For each task, we sample 20 training examples of length $L = 5$ to build the in-context examples, and test on 1 longer sequence of length $2L = 10$. We conduct 20 trials and report the average test accuracy. Table 12 summaries the evaluation results. Overall, except for Copy task where Chat-GPT shows excellent generalization, other tasks are very hard for Chat-GPT, indicating that the length extrapolation problem still poses a big challenge to today AI techniques.

**SCAN**  In this task, we sample 20 examples in the L-cutoff=40 split set (easiest) as in-context learning examples and evaluate on 10 unseen sequences. Chat-GPT totally failed in this task. When testing on the similar length or longer length as the examples, Chat-GPT cannot produce any exact match results (exact match accuracy=0). Below are some failure examples:

- IN: walk and turn opposite right OUT:
  - Chat-GPT output: I_TURN_RIGHT I_TURN_RIGHT I_WALK
  - True output: I_WALK I_TURN_RIGHT I_TURN_RIGHT

- IN: run around left twice and run around right OUT:
  - Chat-GPT output:  I_RUN  I_TURN_LEFT  I_RUN  I_TURN_LEFT  I_RUN I_TURN_RIGHT I_RUN
  - True output: I_TURN_LEFT I_RUN I_TURN_LEFT I_RUN I_TURN_LEFT I_RUN I_TURN_LEFT I_RUN I_TURN_LEFT I_RUN I_TURN_LEFT I_RUN I_TURN_LEFT I_RUN I_TURN_RIGHT I_RUN I_TURN_RIGHT I_RUN I_TURN_RIGHT I_RUN I_TURN_RIGHT I_RUN

| Copy Task | 9(L) | 10(L+1) | 20((L+1)*2) | 40((L+1)*4) | 80((L+1)*8) |
|---|---|---|---|---|---|
| LSTM | 100±0 | 47±0 | 11±0 | 10±0 | 10±0 |
| Location Attention | 100±0 | 93±2 | 51±5 | 28±4 | 20±1 |
| Content Attention | 100±0 | 92±1 | 53±0 | 33±0 | 22±0 |
| Hybrid Attention | 100±0 | 91±1 | 50±1 | 23±3 | 13±0 |
| Transformer | 100±0 | 20±1 | 16±0 | 13±0 | 11±0 |
| NTM | 100±0 | 74±4 | 13±2 | 11±0 | 11±0 |
| DNC | 100±0 | 54±2 | 11±1 | 11±0 | 11±0 |
| Neural Stack | 100±0 | 90±4 | 47±2 | 29±0 | 17±0 |
| PtrNet | 100±0 | 90±2 | 52±1 | 32±1 | 20±0 |
| NRAM | 100±0 | 81±3 | 15±2 | 11±0 | 11±1 |
| ESBN | 100±0 | 92±0 | 34±0 | 11±0 | 11±0 |
| PANM | **100±0** | **100±0** | **84±1** | **52±1** | **36±1** |

Table 13: Copy: accuracy (mean ± std. over 5 runs)

| Reverse Task | 9(L) | 10(L+1) | 20((L+1)*2) | 40((L+1)*4) | 80((L+1)*8) |
|---|---|---|---|---|---|
| LSTM | 100±0 | 96±0 | 53±0 | 33±0 | 22±0 |
| Location Attention | 100±0 | 26±3 | 18±1 | 14±0 | 12±0 |
| Content Attention | 100±0 | 81±25 | 38±11 | 23±4 | 16±2 |
| Hybrid Attention | 100±0 | 98±1 | 50±7 | 24±2 | 15±1 |
| Transformer | 100±0 | 18±0 | 15±3 | 13±1 | 11±0 |
| NTM | 100±0 | 95±7 | 65±27 | 26±13 | 13±1 |
| DNC | 100±0 | 93±3 | 60±18 | 23±6 | 12±1 |
| Neural Stack | 100±0 | 96±1 | 64±4 | 35±3 | 19±1 |
| PtrNet | 100±0 | 77±5 | 32±4 | 22±1 | 12±0 |
| NRAM | 100±0 | 96±1 | 60±3 | 33±2 | 15±2 |
| ESBN | 99±0 | 95±0 | 14±2 | 11±0 | 10±0 |
| PANM | **100±0** | **100±0** | **84±3** | **51±1** | **33±1** |

Table 14: Reverse: accuracy (mean ± std. over 5 runs)

| Mix Task | 9(L) | 10(L+1) | 20((L+1)*2) | 40((L+1)*4) | 80((L+1)*8) |
|---|---|---|---|---|---|
| LSTM | 100±0 | 96±0 | 53±0 | 33±0 | 22±0 |
| Location Attention | 100±0 | 92±10 | 56±1 | 45±0 | 30±6 |
| Content Attention | 100±0 | 61±8 | 57±1 | 14±0 | 12±0 |
| Hybrid Attention | 100±0 | 98±1 | 56±3 | 34±0 | 23±6 |
| Transformer | 100±0 | 18±0 | 15±3 | 13±1 | 11±0 |
| NTM | 100±0 | 95±7 | 65±27 | 26±13 | 13±1 |
| DNC | 100±0 | 91±4 | 58±9 | 19±3 | 11±1 |
| Neural Stack | 100±0 | 87±3 | 50±5 | 14±2 | 11±0 |
| PtrNet | 100±0 | 59±3 | 51±3 | 13±1 | 11±0 |
| NRAM | 99±0 | 82±7 | 48±6 | 17±4 | 10±1 |
| ESBN | 99±0 | 95±0 | 14±2 | 11±0 | 10±0 |
| PANM | **100±0** | **100±0** | **98±1** | **54±0** | **54±1** |

Table 15: Mix: accuracy (mean ± std. over 5 runs)

| Drecall Task | 9(L) | 10(L+1) | 20((L+1)*2) | 40((L+1)*4) | 80((L+1)*8) |
|---|---|---|---|---|---|
| LSTM | 85±7 | 74±16 | 21±2 | 12±1 | 11±0 |
| Location Attention | 88±1 | 82±1 | 30±3 | 19±2 | 13±0 |
| Content Attention | 88±2 | 84±0 | 27±3 | 17±1 | 13±1 |
| Hybrid Attention | 69±25 | 66±24 | 28±4 | 19±2 | 13±1 |
| Transformer | 33±1 | 32±0 | 22±0 | 14±1 | 12±1 |
| NTM | 86±3 | 72±8 | 22±1 | 15±0 | 12±0 |
| DNC | 89±0 | 83±1 | 22±1 | 14±2 | 11±0 |
| Neural Stack | 85±4 | 76±2 | 23±1 | 15±1 | 13±1 |
| PtrNet | 65±14 | 48±7 | 25±6 | 14±1 | 12±1 |
| NRAM | 61±6 | 59±4 | 21±4 | 13±2 | 11±1 |
| ESBN | 90±1 | 86±1 | 22±3 | 11±1 | 10±0 |
| PANM | **92±0** | **89±0** | **45±1** | **22±0** | **16±0** |

Table 16: Drecall: accuracy (mean ± std. over 5 runs)

| PSort Task | 10(L) | 11(L+1) | 21((L+1)*2) | 41((L+1)*4) | 81((L+1)*8) |
|---|---|---|---|---|---|
| LSTM | 87±2 | 83±2 | 28±3 | 16±1 | 12±1 |
| Location Attention | 69±3 | 66±3 | 45±1 | 27±2 | 20±2 |
| Content Attention | **97±0** | 96±0 | 57±6 | 30±7 | 22±5 |
| Hybrid Attention | 85±3 | 81±1 | 33±1 | 25±2 | 23±3 |
| Transformer | 71±9 | 48±8 | 21±3 | 16±4 | 14±4 |
| NTM | 96±2 | 95±3 | 34±18 | 12±1 | 10±0 |
| DNC | 95±0 | 92±2 | 29±7 | 11±1 | 10±0 |
| Neural Stack | 92±2 | 79±2 | 32±3 | 13±2 | 11±1 |
| PtrNet | 77±2 | 71±2 | 43±2 | 24±1 | 19±1 |
| NRAM | 82±3 | 80±2 | 51±2 | 25±1 | 13±1 |
| ESBN | 26±4 | 24±4 | 13±2 | 11±1 | 10±0 |
| PANM | **97±0** | **97±1** | **86±2** | **32±7** | **27±4** |

Table 17: PSort: accuracy (mean ± std. over 5 runs)

| ID Sort Task | 10(L) | 11(L+1) | 21((L+1)*2) | 41((L+1)*4) | 81((L+1)*8) |
|---|---|---|---|---|---|
| LSTM | 48±10 | 40±5 | 20±1 | 13±1 | 11±1 |
| Location Attention | 34±1 | 32±1 | 20±0 | 14±0 | 12±0 |
| Content Attention | 98±1 | 56±1 | 28±2 | 16±0 | 12±0 |
| Hybrid Attention | 32±1 | 31±1 | 19±1 | 14±0 | 12±0 |
| Transformer | 34±2 | 29±0 | 19±0 | 15±0 | 12±0 |
| NTM | 40±23 | 32±17 | 16±4 | 12±2 | 11±0 |
| DNC | 35±1 | 36±1 | 23±2 | 17±2 | 13±0 |
| Neural Stack | 33±3 | 32±1 | 19±1 | 13±1 | 12±0 |
| PtrNet | 27±1 | 24±1 | 15±0 | 12±1 | 11±0 |
| NRAM | 31±2 | 29±1 | 14±0 | 12±0 | 11±0 |
| ESBN | 47±18 | 42±12 | 18±0 | 12±0 | 10±0 |
| PANM | **100±0** | **100±0** | **56±2** | **25±0** | **15±0** |

Table 18: ID Sort: accuracy (mean ± std. over 5 runs)

| Task | add_or_sub | place_value |
|---|---|---|
| U. TRM+ RPE♣ | **0.97 ± 0.01** | 0.75 ± 0.10 |
| TRM + RPE♣ | 0.91 ± 0.03 | - |
| TRM + RPE◇ | 0.91 ± 0.04 | 0 ± 0 |
| TRM♣ | 0.89 ± 0.01 | 0.12 ± 0.07 |
| TRM◇ | 0.86 ± 0.01 | 0.05+0.05 |
| U. TRM♣ | 0.94 ± 0.01 | 0.20 ± 0.02 |
| PANM TRM base (Ours) | 0.91 ± 0.01 | 0.15 ± 0.02 |
| PANM TRM + RPE base (Ours) | **0.97 ± 0.02** | **0.86 ± 0.05** |

Table 19: Mathematics: mean ± std accuracy over 5 runs. ♣ are numbers from Csordás et al. (2021). ◇ is our rerun to confirm the results, which, in some cases, could not match the reported numbers. - means training crash reported in the original papers. We run PANM using the authors' codebase.

---

**Algorithm 1:** PANM training. To simplify, the batch size and number of pointer heads are 1.

---

**Input:** A dataset of sequence pairs $D = \{X_i, Y_i\}_{i=1}^{N_{data}}$, initial $\Phi$ containing the parameters of the Encoder$_\theta$, Pointer Unit PU$_\varphi$ and Controller Ctrl$_\lambda$, $b$ representing the number of bits of the address space, $L_{dec}$ being the maximum number of decoding steps, and function $l$ measuing the length of a sequence.

**Ouput:** $\Phi^*$, trained parameters.

**1** **for** $\{X_i, Y_i\} \sim D$ **do**

    /* Construct the memory           */

**2**    M = Encoder$_\theta(X_i)$

    /* Sample base address. During testing, $a_B$ can be set to 0    */

**3**    $a_B \sim \text{Uniform}\left(\{0,1\}^b\right)$

    /* Generate the address           */

**4**    **for** $j = 0, 1, ..., l(\text{M}) - 1$ **do**

**5**        A$[j] = (a_B + j) \bmod 2^b$

**6**    **end**

    /* Decode with pointers           */

**7**    **for** $t = 0, 1, ..., L_{dec}$ **do**

**8**        Use PU$_\varphi$ and A to compute $p_t^a$ using Eq. 3

**9**        Use M and $p_t^a$ to compute pointer values $^*p_t^a$ (Mode 1) and $^*p_t^c$ (Mode 2) (see §2.3.2)

**10**        Use Ctr$_\lambda$ and pointer values to compute $p_\Phi(y_t|X_i, z_t)$ (see §2.3.3)

**11**        $\hat{y}_t^i = \text{argmax}_{y_t} \ p_\Phi(y_t|X_i, z_t)$

**12**        **if** $\hat{y}_t^i$ *is* EOS **then**

**13**            break

**14**        **end**

**15**    **end**

    /* Compute cross-entropy loss           */

**16**    $\mathcal{L} = -\sum_t \log p_\Phi(y_t = Y_i[t]|X_i, y_{t-}^i)$

**17**    Use $\mathcal{L}$ to update $\Phi$ through backpropagation

**18** **end**

