# OpenReview forum: "Plug, Play, and Generalize: Length Extrapolation with Pointer-Augmented Neural Memory"
_TMLR — Accepted by TMLR_

### Review · Reviewer_2qxe · 2024-08-02

**Summary Of Contributions:**

The paper introduces Pointer-Augmented Neural Memory (PANM), a novel module that employs explicit physical addresses and pointer manipulation techniques to enhance neural networks' capabilities in handling longer sequences and complex symbolic tasks. This approach significantly improves the ability of models to generalize to sequences much longer than those seen during training. PANM demonstrates its versatility and effectiveness across various sequential models, ranging from simple recurrent networks to large language models like Llama2-7B, showing broad applicability.

The experimental results indicate substantial performance gains in critical tasks such as algorithmic reasoning, Dyck language recognition, compositional learning (e.g., SCAN, Mathematical Problems), question answering, and machine translation. PANM's robust memory access mechanisms, including pointer dereferencing and relational access, provide flexible and reliable data retrieval from memory, further supporting enhanced systematic generalization. These contributions collectively address key limitations in current neural network models, particularly in systematic generalization and symbolic processing.

**Audience:**

Yes

**Broader Impact Concerns:**

No.

**Claims And Evidence:**

Yes

**Requested Changes:**

Clarify Terminology: Ensure all technical terms are clearly defined when first introduced. For example, define "<eos>" (end of sequence) in Section 2.1 (and EOSs in the later sections) to avoid any confusion.

Provide More Intuitive Examples: Can you include more intuitive, real-world examples to explain complex concepts, particularly in the sections describing the Pointer Unit and memory access modes (Sections 2.3.1 and 2.3.2)? This will benefit readers grasp the practical implications of the model.

Compare Baselines More Thoroughly: In the Experimental Results section, provide a more detailed comparison with baseline models (e.g., a table comparing key performance metrics (accuracy, precision, recall) for PANM and baseline models across different tasks).

Expand on Limitations: Offer a more detailed discussion on the limitations of PANM, such as computational complexity and potential challenges in implementation.

**Strengths And Weaknesses:**

Strengths:

Innovative Approach: Uses explicit physical addresses and pointer manipulation, improving length extrapolation and systematic generalization.

Versatility: Effective across various models, from simple recurrent networks to large language models.

Performance Gains: Significant improvements in algorithmic reasoning, language recognition, compositional learning, and NLP tasks.

Robust Memory Access: Implements two robust memory access modes for flexible and reliable data retrieval.


Weaknesses:

Complexity: Increased implementation and computational complexity due to explicit pointer operations and memory management.

Resource Intensive: Higher computational demands may limit applicability in resource-constrained environments.

Task Specificity: While excellent for symbolic and length extrapolation tasks, the benefits for other task types may be less pronounced.

---

> ### Author Response · Authors · 2024-08-20
> **Response to Reviewer 2qxe (1/2)**
>
> Thank you for your feedback. We'll address your concerns point by point below.
> - Weaknesses:
>     + **Complexity, Resource Intensive**: We acknowledge that augmenting models with PANM introduces some computational and memory overhead. However, as detailed in Appendix D, the increase in computational complexity is relatively modest—only about 20-30% for smaller models like LSTM. For larger models, such as LLMs, the impact is minimal; in fact, LLM+PANM operates nearly as efficiently as LLM alone because PANM's operations are negligible compared to the overall computation of an LLM. Similarly, the memory requirements for PANM are minimal. The only significant addition is the Address Bank, with each slot being just 10-dimensional (10 bits). The parameters for the Pointer Unit and Controller are also minor, especially when compared to the memory demands of large backbone models.
>    + **Task Specificity**: While PANM's strengths are most evident in symbolic and length extrapolation tasks, these areas are critical and pervasive in practical applications. They also represent a significant bottleneck in current deep learning. Enhancing performance in these tasks addresses a core challenge and brings substantial value to a wide range of real-world problems
> - Requested Changes:
>    + **Clarify Terminology**: Thank you for pointing that out. We have updated the notation consistently using 'EOS', and added its definition in Section 2.1 of the revised version.
>    + **Provide More Intuitive Examples**:  We have created several examples to clarify the concepts in our paper and added them briefly across Sec. 2.3 of the revision. The examples are:
>       - Pointer Unit: Consider a task where you need to summarize a lengthy article by extracting key points from specific paragraphs. Traditional methods may struggle with long documents, but PANM's Pointer Unit (PU) functions like a sophisticated "bookmark" system. It begins at the targeted paragraph and sequentially progresses to each key point, effectively managing documents of any length by using relative positions instead of content.
>       - Mode 1: Direct Content Retrieval. Consider a task where you need to extract specific sentences from a document based on their position, like retrieving sentences from a given paragraph in a text corpus. In Mode 1, PANM’s pointer dereferences directly, similar to using a precise index to access specific parts of the text. For instance, if the Pointer Unit (PU) points to sentence 12, it directly retrieves that sentence from the document. This is like accessing a particular row in a data table, regardless of the document’s length.
>        - Mode 2: Complex Reasoning and Relationships.  Imagine you need to summarize a document by identifying the most important sentences based on their relevance to a query. In Mode 2, instead of directly retrieving sentences, the PU uses the content of a sentence to generate a query that scores and ranks sentences based on their relevance. This process is akin to running a search query that evaluates and selects the most pertinent sentences from the entire document, based on contextual understanding and relevance to the query.
>    + **Compare Baselines More Thoroughly**: Thank you for your comment. Given the variety of tasks we examine, each with its own set of subtasks, baselines, and metrics, it’s challenging to consolidate everything into a single table. Instead, in Sec. 3, we have provided detailed tables for each task. However, to address your suggestion, we have summarized the results into a single table by averaging the outcomes of subtasks and tasks that share the same baselines and metrics. The summarized results, now included in Table 2 of the revision, (attached below) demonstrate that PANM consistently outperforms the best baselines on average across different tasks.
> | Task                      | Key Metric             | Best Baseline         | PANM (Ours) |
> |---------------------------|------------------------|-----------------------|-------------|
> | Algorithmic Reasoning      | ↑ Exact Match (%)      | 56.3 (Other Max)       | **68.2**        |
> | Dyck Language Recognition  | ↑ Exact Match (%)      | 28.2 (SRNN)            | **70.0**        |
> | Compositional Learning     | ↑ Exact Match (%)      | 82.6 (U. TRM+RPE)      | **88.6**        |
> | bAbI QA                    | ↑ Accuracy (%)         | 77.5 (TRM)             | **83.0**        |
> | SQUAD QA                   | ↑ F1 (%)               | 75.0 (BERT)            | **77.0**        |
> | Machine Translation        | ↓ PPL                  | 42.7 (TRM)             | **32.7**        |
> | LLM Algorithmic Tasks      | ↑ BLEU (%)             | 14.7 (Llama)           | **62.7**        |
> | BIG-bench Tasks            | ↑ Exact Match (%)      | 11.5 (Llama)           | **22.5**        |

---

> > ### Author Response · Authors · 2024-08-20
> > **Response to Reviewer 2qxe (2/2)**
> >
> > + **"Expand on Limitations"**:  Following the previous discussion on the complexity of PANM, we have expanded the Conclusion section with the limitations in the revision as follows:
> >        > **Computational Complexity:** PANM introduces additional computational overhead, particularly when used with smaller models like LSTMs, where complexity can increase by 20-30%. For larger models, such as LLMs, this overhead is minimal, making PANM’s operations relatively inexpensive. **Memory Requirements:** The additional memory needed for the Address Bank and Pointer Unit is small relative to large backbone models. However, these components do contribute to overall memory usage, which may be a consideration in resource-constrained environments. **Implementation Challenges:** Integrating PANM into existing architectures might require engineering effort. Ensuring seamless interaction between the backbone and PNAM and tuning PANM hyperparameters for optimal performance on specific tasks may require additional experimentation.
> > Despite these limitations, we believe that PANM’s ability to address symbolic and length extrapolation challenges—key bottlenecks in deep learning—makes it a valuable tool in many practical applications.

---

### Review · Reviewer_fkQp · 2024-08-13

**Summary Of Contributions:**

In this paper, the authors introduce PANM, which is an external memory module that endows a neural network with memory, allowing for generalization. Crucially, the approach relies on the pointer and memory being separate, as opposed to previous works, where the pointer and memory are the same.

**Audience:**

Yes

**Claims And Evidence:**

Yes

**Requested Changes:**

1. Notation changes to make paper more readable (required)
2. Baselines against other memory-augmented NNs.

**Strengths And Weaknesses:**

# Strength

To me the biggest strength is the intuitiveness and simplicity of the idea. I love the inspiration in programming languages which really motivates the approach. Moreover, the training is extremely simple, which is in contrast to previous approaches that rely on using variational inference for training the memory module (as a posterior over the memory contents is inferred from the training data).

# Weaknesses

For me the biggest weakness is in the writing. I absolutely love the motivation and sets up the approach perfectly, but it is a little hard to gather what is going on. For starters, I think a little more hand-holding could be done in the beginning of section 2.2; for instance, I think spelling out the copy-list example would be great. I also think the choice of notation isn't helping here. $p$ (and $p_E$) is a specific pointer but $p_b$ corresponds to the size of the address bank.

Next, while the experimental results clearly demonstrate that PANM outperforms non-memory augmented neural networks, I think comparisons against other memory-augmented NNs are needed, especially given the motivation that the use of PANM specifically allows for the networks to generalize.

---

> ### Author Response · Authors · 2024-08-20
> **Response to Reviewer fkQp**
>
> We appreciate your feedback and will address your concerns point by point below.
> - Weakness:
>   + **"For me the biggest ..."**: Thank you for your suggestion. We have revised Section 2.2 to add more guiding explanation for the copy list example as suggested:
> >We can manipulate the pointer to execute various tasks. For example, given a list $X$ storing elements in consecutive memory slots, $\\&X$ denotes the pointer of the first element of the list. If the task is to copy the list $X$ to a new list $Y$, using pointers, this task can be executed by iterating over the elements of the list and copying each element from $X$ to $Y$ regardless of the list length and the values in $X$. The copying process can be described as follows: (1, assignment) Initialize 2 pointers pointing at the first element of $X$ and $Y$, respectively: $p_{X}=\\&X;p_{Y}=\\&Y$; (2, dereference) Access the value pointed to by $p_{X}$ (i.e., the current element of $X$) and copy it to the location pointed to by $p_{Y}$ (i.e., the corresponding position in $Y$): $*p_{Y}=*p_{X}$; (3, arithmetic) Increment both pointers to move to the next elements in their respective lists, $p_{X}=p_{X}+1;p_{Y}=p_{Y}+1$. Repeat this procedure until all elements in $X$ have been copied to $Y$.
> - Requested Changes:
>   + **"1. Notation changes ..."** Thank you for the comment. We recognize that the notations $p_B$ and $p_E$, representing the base and end addresses, could be easily confused with an arbitrary pointer $p$. To address this, we have updated the notations to $a_B$ and $a_E$, respectively, to prevent any ambiguity. We hope this revision clarifies your concern.
>   + **"2. Baselines against ..."** We would like to clarify that we have compared our methods with various memory-augmented NNs (MANN) across our experiments. For example, in Sec. 3.1 the MANN baselines are NTM, DNC, NRAM, NeuralStack, and ESBN. We described these baselines in Sec. 3.1 (Baseline paragraph). In Sec. 3.2, the MANN baseline is SRNN, a state-of-the-art MANN on Dyck language recognition.  In other sections, Transformer variants are the main baselines, which can be regarded as a form of memory-augmented neural network [1]. Therefore, across tasks, our main baselines are MANNs, not non-memory augmented neural networks as you mentioned.
> That said, there are other kinds of MANNs that we have not covered. For example, we did not include the variational MANN approaches that you mentioned as the baselines because:
>      - As you pointed out, they are complicated and not easy to train
>      - They are more suitable for image-generation tasks [2,3], which is out of this paper's scope
>
> In conclusion, we believe that we carefully selected MANN baselines that are both reasonable and well-established for the length extrapolation tasks under study.
>
> [1] Katharopoulos, Angelos, Apoorv Vyas, Nikolaos Pappas, and François Fleuret. "Transformers are rnns: Fast autoregressive transformers with linear attention." In International conference on machine learning, pp. 5156-5165. PMLR, 2020.
>
> [2] Wu, Yan, Greg Wayne, Alex Graves, and Timothy Lillicrap. "The Kanerva Machine: A Generative Distributed Memory." In International Conference on Learning Representations. 2018.
>
> [3] Pham, Kha, Hung Le, Man Ngo, Truyen Tran, Bao Ho, and Svetha Venkatesh. "Generative pseudo-inverse memory." In International Conference on Learning Representations. 2022.

---

> > ### Comment · Reviewer_fkQp · 2024-08-21
> > **Response**
> >
> > Thank you for the updates! I think most of my concerns have been addressed!

---

### Review · Reviewer_GYiy · 2024-08-13

**Summary Of Contributions:**

The authors propose Pointer-Augmented Neural Memory, which can be integrated into a wide range of neural networks in order to improve generalisation on symbol processing tasks, improve performance of compositional learning and improve scalability and length extrapolation. Their approach is inspired by the use of pointers in computing and as a part of their model architecture, they train networks that learn to manipulate addresses (which refer to the elements of the encoded input sequence) in order to solve the task at hand.

Each element in the input sequence is encoded and serves as data in the memory bank. Each element in the memory (representations of input sequence elements) is assigned a physical address in the address bank, starting from the base address which is uniformly sampled from the address space (such that all possible address ranges are observed during training, avoiding fitting the model based on the values of the addresses in themselves). These addresses are used in subsequent pointer arithmetic in the decoding process. This data in the memory bank, i.e. the encodings of the input sequence elements, is not further altered in the decoding process and is not used to compute the pointers, reflecting the stated principle of manipulating the pointers in a way that is decoupled from input data.

In each iteration, a GRU computes a new hidden state based on the previous hidden state and the previous pointer. The alignments between the new hidden state and the outputs of the neural network that transforms each of the entries of the dress bank into an equal-dimension space are softmaxed - resulting in a vector of weights for each element in the address bank. The weights are used for computing a weighted sum of the memory addresses in the bank - resulting in a new memory address which represents a combination of physical addresses and which is independent of the length of the sequence. The computed pointer is used in the subsequent iteration of the GRU.

In the Addressing Mode 1, the pointer obtained from the PU is dereferenced by using the same aforementioned weights to compute a weighted sum of entries in the memory data. While the aspects of Mode 1 are very similar to the classic attention mechanism in the sense that a weighted sum of the encoded input sequence elements is eventually computed, the fact that the weights are computed based on the corresponding addresses in a way that is decoupled from the input data, to the best of my knowledge, makes this approach novel. The pointer unit does not receive the input sequence as input, but learns to manipulate the addresses in a way that it can serve appropriate data for the decoding process. The authors argue that this results in particularly high performance on input sequences that are longer than those observed in training data, which is subsequently confirmed in the experiments.

The Addressing Mode 2 on the other hand, uses an attention mechanism with the data memory serving as both keys and values, while the pointers computed by the PU are transformed into a query vector using an additional neural network, resulting in an additional pointer value that captures the most relevant context from the data memory for the decoder, which has proven itself as a critical component in content-based tasks that require the token value as well. In each decoding iteration, the Mode 1 and Mode 2 pointer values are concatenated and finally given as input to the decoder.

**Audience:**

Yes

**Claims And Evidence:**

Yes

**Requested Changes:**

- From the sentence "$h^a_0$ is initialized as 0 and $1 ≤ n ≤ l(X)$, $\varphi$ denotes the parameter of the PU..." in 2.3.1., it is still not clear how $h^a_0$ is initialised - is it just an all-zero vector and the statement $1 ≤ n ≤ l(X)$ should have been separate?
- $\varphi $ denotes the parameter of the PU in 2.3.1. - I assume there are more than one parameters and that $\varphi$ indicates the parameters of the PU in general?
- Since the output of the PU ($p^a_t$) is a combination of addresses in the address bank, and not one of the addresses itself, I find the Figure 1 confusing. To my understanding, the $*p^a_t$ will not necessarily (or nearly ever) match any of the entries of the data memory, while in the figure, both the output of the PU and one of the entries in the data memory are indicated as $*p^a_t$ - I believe clarifying this would be important and I'd encourage the authors to potentially correct my understanding or otherwise address the potential confusion in what is otherwise a prominent figure in the paper
- In Figure 3. - Where the authors show the perplexity on the Multi30K and report "The best test perplexity over 2 settings for different number of Transformer’s layers", the perplexity increases sharply already with the addition of the second layer. Given the degree of overfitting that is observed, I'd appreciate a comment on whether their choice of the Transformer architecture could have been in any way disadvantageous for the baseline, and whether simpler techniques like early stopping could have potentially been used to erase the performance gap to a degree.

**Strengths And Weaknesses:**

Strengths:

- The manuscript is generally well written and easy to follow, and is accompanied by an extensive appendix featuring additional experiments and ablation studies
- Well justified choice of different symbolic reasoning tasks, as well as the baselines.
- Significant improvements over baselines, particularly drastic on test sequences that are significantly longer than the ones observed in the training data, clearly addressing the main stated objective of the work.

Weaknesses:
- While the focus on test sequences longer than the ones observed during training is definitely welcome, it would also be interesting to see performance on test sequences that are shorter than the largest length observed during training. It would be interesting to see if significant performance improvements emerge on those kinds of sequences as well, as in most practical settings, most test sequences would not be longer than the longest train sequence, and a positive result there would point to an even more wide-ranging benefit of using the proposed approach.
- While I understand the focus on demonstrating the improvements to generalization on a wide range of settings as opposed to outperforming the state-of-the-art on a select few tasks, due to the fact that in the experiments the authors considered highly specific evaluation scenarios with customized train-test splits, it is somewhat difficult to gauge the usefulness of their approach in a more general setting, or compare their approach to other methods aimed at improving systematic generalization.  For instance, due to the previously mentioned focus on longer test sequences required a hand-crafted evaluation setup on multiple tasks, making direct comparison with other works more difficult.

---

> ### Author Response · Authors · 2024-08-20
> **Response to Reviewer GYiy**
>
> Thank you for your review. We have addressed each of your concerns in detail below.
> - Weaknesses:
>   + **"While the focus on ..."**: Thank you for your suggestion. While short-to-long generalization is more common in the literature due to the ease of collecting and training on shorter sequences, our pointer mechanisms are not limited to this setting. To demonstrate this, we conducted additional experiments on long-to-short generalization using a synthetic task (Mix) and a practical task (Machine Translation). The results, now included in the revised paper (Appendix D.6), show that our method significantly outperforms the best baselines in these tests. Specifically, in Mix, PANM achieves an improvement margin of at least 14% accuracy compared to Location Attention, the best baseline in this task. In Machine Translation, the best baselineTransformer model overfits early. In contrast, PANM achieves reasonable perplexity on the test data, outperforming the best Transformer result by approximately 25 perplexity points.
>   + **"While I understand the focus ..."**: Thank you for your thoughtful comment. Because we aim to show the versatility of our method, we choose diverse test suits, where in many cases, we have to modify dataset splits to make them suitable for length extrapolation tests. That said, we would like to note that we dedicated section 3.3 to evaluating our method on common length extrapolation benchmarks (Csordás et al., 2021). The baselines, evaluation protocols, and dataset splits are standardized by earlier works.
>
> - Requested Changes:
>    + **"From the sentence ..."**: yes, $h^a_0$ is all-zero vector and the statement 1≤n≤l(X) is for explaining the index n in Eq. 2. We have revised to separate them as follows, "where $h_{0}^{a}$ is initialized as $\overrightarrow{0}$ in Eq. (1). In Eq. (2), $1\leq n\leq l(X)$ ..."
>    + **"$\varphi$ denotes the parameter ..."**: yes, $\varphi$ is a general notation for parameters of the neural network  $g^a$. We have revised the text, from "parameter" to "parameters" to clarify it.
>    + **"Since the output of ..."**: You are correct; $p_t^a$ represents a mixture of addresses, and the figure is meant for illustrative purposes. It specifically shows a special case where $p_t^a$  exactly matches one address in the address bank. To address any confusion, we have updated the caption to clarify: "We note that for illustration purposes, the figure depicts a special case where the pointer perfectly matches an address (e.g., $p_{t}^{a}=0010$), in practice, the pointer may not point exactly to a single address."
>   + **"In Figure 3. ..."**: In this task, we use the Transformer model as the baseline since it is a standard architecture for machine translation. We enhanced the Transformer with our PANM memory while maintaining the same backbone architecture to ensure a fair comparison, without introducing any disadvantages. For this experiment, we follow the practice in  Csordás et al. (2021) training with a fixed number of epochs (60). The number of epochs is chosen such that the testing loss reaches a plateau and likely cannot improve further, which can be seen as a form of early stopping (see Fig. 3b (right)). Overall, the curves show that the performance gap between our method and the baseline is consistently maintained throughout the training epochs.

---

### Author Response · Authors · 2024-08-20
**General Response and Summary of Changes**

We sincerely appreciate the Reviewers' constructive and valuable feedback on our paper. We are pleased that they found our paper " well written and easy to follow" (Reviewer GYiy), and recognized the "intuitiveness and simplicity" (Reviewer fkQp) and "innovation and versatility" (Reviewer 2qxe) of our method. We are also encouraged by the positive feedback on our experiments, which were noted for their "justified choice of different symbolic reasoning tasks, as well as the baselines", "clearly addressing the main stated objective of the work" (Reviewer GYiy) and "significant improvements in algorithmic reasoning, language recognition, compositional learning, and NLP tasks" (Reviewer 2qxe).
We have addressed the remaining concerns in the individual reviews and made the following updates to the manuscript:
- Conducted additional experiments on long-to-short generalization.
- Enhanced the clarity of notation and figures.
- Provided more detailed descriptions and guiding examples, including the copy-list motivation, the operation of the Pointer Unit, and the two Modes of Access.
- Included a comprehensive table comparing PANM with other baselines across tasks and metrics.
- Added a detailed discussion on potential limitations, including computation, memory, complexity, and implementation.

---

### Decision · Action_Editor_SPZm · 2024-09-13

**Recommendation:** Accept as is

**Comment:**

The paper addresses an important problem, takes a novel approach, and provides substantial evidence to substantiate its claims. The reviewers originally had some concerns regarding the experimental setup and the writing, but these have been largely addressed in the revision.

**Audience:**

The paper targets a fundamental problem in contemporary machine learning and presents an interesting new method for addressing it. I believe the results here should interest a nontrivial part of the TMLR community.

**Claims And Evidence:**

The paper presents pointer-augmented neural memory (PANM), a differentiable module whose use, the authors claim, enhances the ability of sequence-processing neural networks to extrapolate to longer sequences. To validate this claim, the paper augments RNNs and Transformers with PANMs and evaluates the resulting models on a range of sequence-processing tasks. It is shown that the augmentation leads to substantially better performance over baselines at several of these tasks.